# Multimodal monitoring of human cortical organoids implanted in mice reveal functional connection with visual cortex

Madison N. Wilson [1,19], Martin Thunemann [2,19], Xin Liu [1], Yichen Lu [1], Francesca Puppo[3], Jason W. Adams[3,4], Jeong-Hoon Kim[1], Mehrdad Ramezani[1], Donald P. Pizzo[5], Srdjan Djurovic [6,7,8,9], Ole A. Andreassen [7,9,10,11,12], Abed AlFatah Mansour [13,14], Fred H. Gage [13], Alysson R. Muotri[3,4,15,16,17], Anna Devor [2,18,20] ✉ & Duygu Kuzum [1,20] ✉

Human cortical organoids, three-dimensional neuronal cultures, are emerging as powerful tools to study brain development and dysfunction. However, whether organoids can functionally connect to a sensory network in vivo has yet to be demonstrated. Here, we combine transparent microelectrode arrays and two-photon imaging for longitudinal, multimodal monitoring of human cortical organoids transplanted into the retrosplenial cortex of adult mice. Two-photon imaging shows vascularization of the transplanted organoid. Visual stimuli evoke electrophysiological responses in the organoid, matching the responses from the surrounding cortex. Increases in multi-unit activity (MUA) and gamma power and phase locking of stimulus-evoked MUA with slow oscillations indicate functional integration between the organoid and the host brain. Immunostaining confirms the presence of human-mouse synapses. Implantation of transparent microelectrodes with organoids serves as a versatile in vivo platform for comprehensive evaluation of the development, maturation, and functional integration of human neuronal networks within the mouse brain.

Recent progress in stem cell technology has yielded human cortical organoids, three-dimensional neuronal networks derived from differentiated human induced pluripotent stem cells (hiPSCs) to recapitulate some features of cerebral cortex. This bioengineering advance shows promise as the next generation of disease models, platforms for drug screening and personalized medicine, and transplantable neural prosthetics to restore specific lost, degenerated, or damaged brain regions[1,2]. Previous studies showed that cortical organoids in a dish

[1]Department of Electrical and Computer Engineering, University of California San Diego, La Jolla, CA, USA. [2]Department of Biomedical Engineering, Boston University, Boston, MA, USA. [3]Department of Pediatrics, University of California San Diego, School of Medicine, La Jolla, CA, USA. [4]Department of Cellular and Molecular Medicine, University of California San Diego, School of Medicine, La Jolla, CA, USA. [5]Department of Pathology, University of California San Diego, La Jolla, CA, USA. [6]Department of Medical Genetics, Oslo University Hospital, Oslo, Norway. [7]NORMENT Center, Oslo, Norway. [8]Department of Clinical Science, University of Bergen, Bergen, Norway. [9]K. G. Jebsen Center for Neurodevelopmental Disorders, University of Oslo, Oslo, Norway. [10]Division of Mental Health and Addiction, Oslo University Hospital, Oslo, Norway. [11]Institute of Clinical Medicine, University of Oslo, Oslo, Norway. [12]Oslo University Hospital, University of Oslo, Oslo, Norway. [13]Laboratory of Genetics, The Salk Institute for Biological Studies, La Jolla, CA, USA. [14]Department of Medical Neurobiology, The Hebrew University of Jerusalem, Ein Kerem-Jerusalem, Israel. [15]Center for Academic Research and Training in Anthropogeny, University of California San Diego, La Jolla, CA, USA. [16]Archealization Center, University of California San Diego, La Jolla, CA, USA. [17]Kavli Institute for Brain and Mind, University of California San Diego, La Jolla, CA, USA. [18]Athinoula A. Martinos Center for Biomedical Imaging, Department of Radiology, Harvard Medical School, Massachusetts General Hospital, Charlestown, MA, USA. [19]These authors contributed equally: Madison N. Wilson, Martin Thunemann. [20]These authors jointly supervised this work: Anna Devor, Duygu Kuzum. ✉e-mail: adevor@bu.edu; dkuzum@eng.ucsd.edu

have expression profiles matching that of fetal and postnatal human cortical development[3] and that organoids could produce electrophysiological network activity resembling that of neonatal human brain electroencephalography[4]. Compared to two-dimensional neuronal cultures, cortical organoids exhibit complex features of neuronal organization such as cortical layers[5,6] and greater cell-type diversity[1,7]. Compared to animal models, cortical organoids preserve the genetic background of individual human patients[1,8]. Furthermore, recent work has reported that organoids implanted in vivo in mouse cortex can establish functional vascularization of the transplant to provide nutrients and oxygen supply, preventing the necrotic cell death in the organoid core[9], and extend axonal projections into the surrounding host tissue[10].

Although these promising studies have laid the foundation for organoid research in vivo, they largely focused on local and acute measurements of neural activity from a small number of cells. Chronic functional responses to external sensory stimuli generated by transplanted organoids into adult recipients have yet to be demonstrated. This would require longitudinal studies focusing on both the structural and electrophysiological development of transplanted organoids. Existing recording technologies currently limit the feasibility of such multimodal experiments. To address that technological gap, we developed an experimental paradigm based on optically transparent graphene microelectrode technology[11,12] for longitudinal, multimodal monitoring of the development, maturation, and functional integration of human neuronal networks chronically implanted within the retrosplenial cortex in the mouse brain. Using awake mice with chronic implants within the retrosplenial cortex, we demonstrated longitudinal recordings of electrical responses to visual stimulation from

cortical organoids and surrounding tissue along with optical imaging of vascularization into the organoids. These longitudinal multimodal recordings allowed us to study neural activity propagation across the organoid/cortex boundary, functional response to sensory stimuli generated by the organoid, and modulation of spiking activity of the cells in the transplanted organoid by the neural activity in the surrounding cortex. We performed post-mortem histological analysis to study morphological integration and synaptic connection of human organoids with the surrounding mouse cortex. Furthermore, we extended this paradigm to study the differences in response to anesthesia between implanted organoid and host cortex. Our work provides a unique multimodal approach for studying the evolution of organoid activity and its functional integration with surrounding cortex during its maturation in vivo. Our work will pave the way for future studies of functionally integrated human organoid transplants focusing on a wide range of scientific questions and potential clinical use of organoids.

## Results

### Organoid and microelectrode array co-implantation allow longitudinal monitoring of organoid graft

The optical transparency of graphene microelectrodes enables seamless integration of electrical recording with multiphoton imaging and optogenetic stimulation[11]. We used transparent graphene microelectrode arrays to combine electrical recordings with optical imaging of xenografted cortical organoids and surrounding host cortex. The microelectrode arrays have sixteen 100 μm electrode pads spaced 500 μm apart, covering an area of 2 mm × 2 mm (Fig. 1a). Before implantation, we performed electrochemical impedance spectroscopy

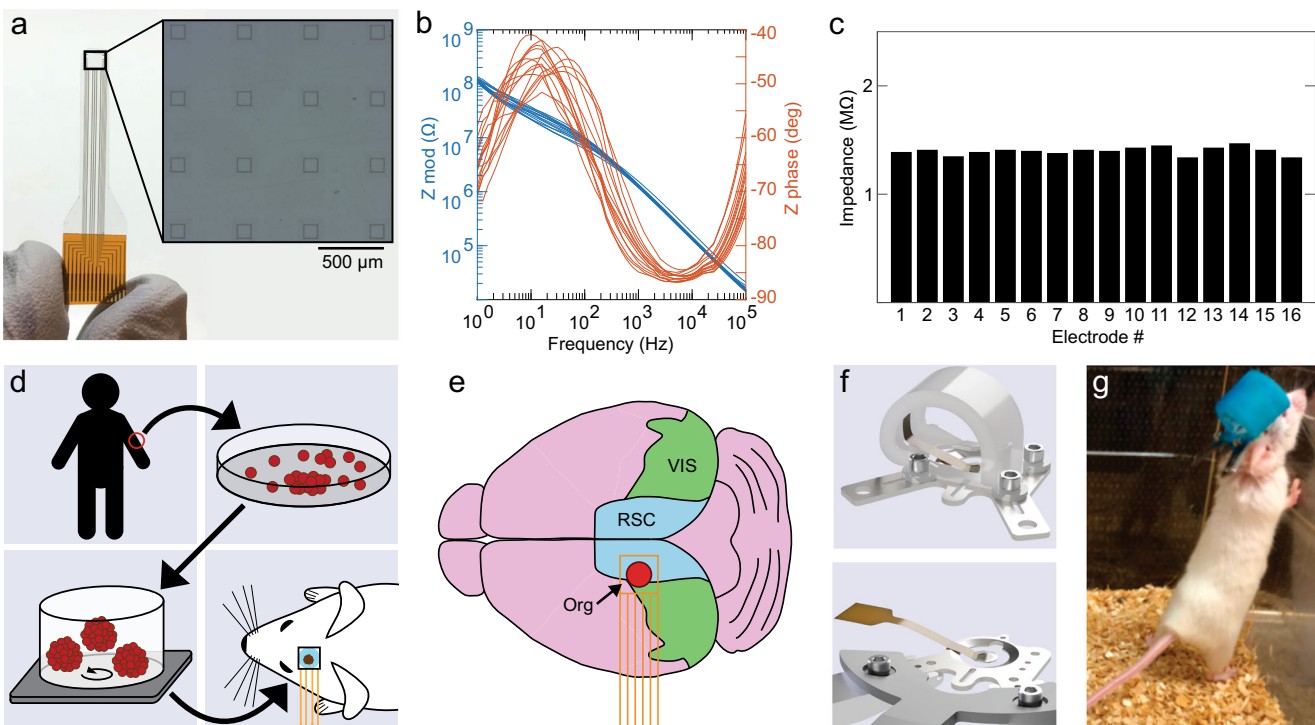

**Fig. 1 | Generation of organoids and co-implantation with microelectrodes in mouse cortex. a** Transparent 16-channel graphene microelectrode array, representative of the eight arrays implanted in eight animals. The inset shows a high-magnification image of the active electrode pads. **b** Electrochemical impedance spectroscopy of a representative graphene microelectrode array; traces represent individual channels of the same array. **c** Impedance of a representative array measured at 1 kHz before implantation. Source data for **b** and **c** are provided as a Source data file. **d** Generation of human induced pluripotent stem cells (hiPSCs) from skin fibroblasts; culture of hiPSC-derived organoid cultures for implantation into mouse cortex. **e** Location of organoid (Org) implantation site and position of microelectrode array (orange) in relation to retrosplenial (RSC) and visual (VIS) cortex. **f** Rendering of headpost with protective cap installed (top), and of headpost fixed to the holder for recording with protective cap removed (bottom). **g** Mouse with headpost and protective cap in its home cage.

(Fig. 1b, c); average impedance magnitudes were around 1.4 MΩ at 1 kHz. For implantation, we bonded the backside of the microarray to a glass plug consisting of two 3-mm coverslips and one 5-mm coverslip. The workflow for generation and implantation of cortical organoids is outlined in Fig. 1d. hiPSCs were differentiated and aggregated into three-dimensional cultures following a previously devised protocol[4]. Seven to nine weeks after generation, single cortical organoids were chosen for implantation based on size and shape. Following our previous work[9], we targeted the left retrosplenial cortex (RSC) as the implantation site. A dense vascular network underlies the RSC[9], which increases the likelihood of vascularization of the xenografted organoid. We used adult (8- to 12-week-old) NOD/SCID mice as recipients. After installation of a titanium headpost, we removed a circa 3 × 3 mm piece of skull and dura mater over the target region. We aspirated a circa 1-mm³ volume of cortex while avoiding large arteries and veins, placed a single cortical organoid into the void, and placed the graphene microelectrode array/glass assembly on top to close the exposure (Fig. 1e). Array wires were protected with a custom 3D-printed cap (Fig. 1f, top) while the animal was in its home cage (Fig. 1g); the cap was removed during recording sessions (Fig. 1f, bottom). After a recovery period of 1 week, electrophysiological recordings were conducted in six mice every 2 weeks for a total of 8 to 11 weeks. Most microelectrode impedances remained stable over the study period (Supplementary Fig. 1). Figure 2a shows the exposure of a representative animal 69 days after implantation; squares mark graphene electrode pads. The xenograft is outlined in red, showing slight recession and discoloration compared to the surrounding cortex.

## Organoids generate neural responses to sensory stimuli

We first employed our longitudinal multimodal monitoring approach to investigate whether organoids could generate a functional response to sensory stimuli by applying a visual stimulus and recording the subsequent electrical responses. We hypothesized that the organoids would begin to establish functional activity to visual stimuli as they integrated with the visual cortex host tissue. Stimulus-induced electrical responses were elicited using light pulses from an optical fiber-coupled white-light LED placed in front of the right eye of the animal. Over the experiment duration, we observed that local field potentials (LFP) did in fact appear in electrode channels above the organoids in recordings taken 3 weeks after organoid implantation until experiment termination (n = 6 animals, Supplementary Fig. 2). Figure 2 shows representative results from one experiment 69 days after organoid implantation in mouse cortex. Figure 2a shows a brightfield image of the implantation site. Six electrode channels over the implanted organoid are marked in red. Among the channels over the surrounding cortex, the ones in closest proximity to visual cortex are marked in blue. The remaining electrodes (black) overlay tissue close to the border between cortex and superior sagittal and transversal sinuses. Figure 2b–e shows recording results (low-pass filtered, 250 Hz) from a session where 100-ms light pulses were delivered at 2 Hz for 4 s. The low noise of graphene electrodes enabled reliable single trial recordings (Fig. 2b). LFPs followed a consistent biphasic shape at the onset of each light pulse[13,14]. The channels overlaying the organoid (red) exhibited visual stimulation response, while the strongest responses were measured in (blue) channels in close vicinity to visual cortex. Average LFP responses to light stimulation for all 16 channels is shown in Fig. 2c. LFPs exhibit continuity across cortex/organoid border showing a propagation pattern starting at the area closest to visual cortex and expanding to the implanted organoid (Fig. 2d). LFP responses of the organoid matching that of the surrounding cortex and uninterrupted propagation of LFPs from the cortex to the organoid suggest functional connectivity at the time of the recording. The LFPs in organoid channels were strongest in later recording sessions (>50 dpi), as shown in Fig. 2, and emerged around 3 weeks post-implantation when the first recordings that included light stimuli were taken (Supplementary Fig. 2).

Broadband LFP signals (low-pass filtered, 250 Hz) are the spatial summation of extracellular potentials generated by several mechanisms including synaptic transmission, action potentials, intrinsic currents and volume conduction. Relative contribution of these mechanism and spatial locality and specificity of LFPs depend on the frequency. Volume conduction refers to the currents flowing in the tissues surrounding active neuronal sources and its effects are considered instantaneous across channels[15–17]. Therefore, to eliminate the probable volume conducted signal shared between channels, we performed independent component analysis (ICA) during our data preprocessing. ICA is a method that statistically performs blind source separation of the common signals contributing to channel recordings, such as volume conducted signals[16,17]. By manually removing the signal components shared across all channels, we ensured that noise and artifacts due to volume conduction (and other environmental factors such as electrical noise and mouse motion) were removed before signal analysis. Going further, electrophysiological signal traveling between cortical regions will have a time delay based on underlying axonal projections[18]. Therefore, to test whether LFP responses were locally generated by the organoid as a result of biological signals propagating via synaptic connections from the cortex or they were signals detected as a result of volume conduction, we examined the relative time-course of LFPs across recording sites. We observed a propagation pattern starting at the area closest to visual cortex (bottom right) and expanding to the implanted organoid area. We quantified this propagation towards the organoid region using the amplitude and delay of the first LFP peak of the trial average (Fig. 2d). Peak LFP amplitudes for the channels overlapping with the visual cortex were around 200 μV and occurred 36 ms after stimulation onset. The channels overlapping with the organoids also detected LFPs with amplitudes of ~50 μV that occurred 41.7 ms after stimulus onset. The 36 ms delay between the stimulation onset and LFP responses measured from the visual cortex channels matches previous observations of evoked response latencies in intact mouse visual cortex[14,19,20]. The 5.7 ms delay we observed between the furthest organoid and cortical channels (p < 0.01, Student's one-tailed t test, Supplementary Fig. 3) is on the order of latencies observed between the intact visual cortex and the RSC region in the literature[21]. The consistency in our observed delay times from the visual cortex to the organoid with the literature delay times between the visual cortex and RSC suggests functional connectivity and propagation between the host and transplanted organoid.

Spectral analysis with Morlet wavelets revealed increases in signal power at several frequencies during visual stimulation (Fig. 2e) for both cortex and organoid channels. First, we observed increased signal power at the stimulation frequency and its second harmonic, consistent with previous functional activity described in intact cortices of mice[14], cats[22], and humans[23,24] (Supplementary Fig. 4). Second, visual light stimulation resulted in increases in gamma (30–150 Hz) power, which is consistent with previous work performed in intact visual cortex of mice in vivo[25,26]. Specifically, we observed an increase in power at 60–100 Hz, which coincided with the onset of individual stimuli. Notably, channels overlapping the implanted organoid also showed a gamma response, which is considered a more localized measure due to attenuated propagation of high-frequency signals in brain tissue[27] (Fig. 2e). LFP in the high gamma frequency range (>100 Hz) has been suggested to be strongly correlated with local spiking activity[27–32]. In studies with macaque monkeys, high gamma power of LFPs was found to be highly correlated with spiking activity of neurons[31,33]. Our recordings also showed high gamma range activity in both organoid and cortex channels (Fig. 2e) suggesting that the activity originated from the underlying organoid or cortical tissue and not volume conducted signals. Overall, responses to visual stimulation were clearly detectable in channels overlaying the organoid (Fig. 2e, channel 1) as well as the cortex channels close to visual cortex (Fig. 2e,

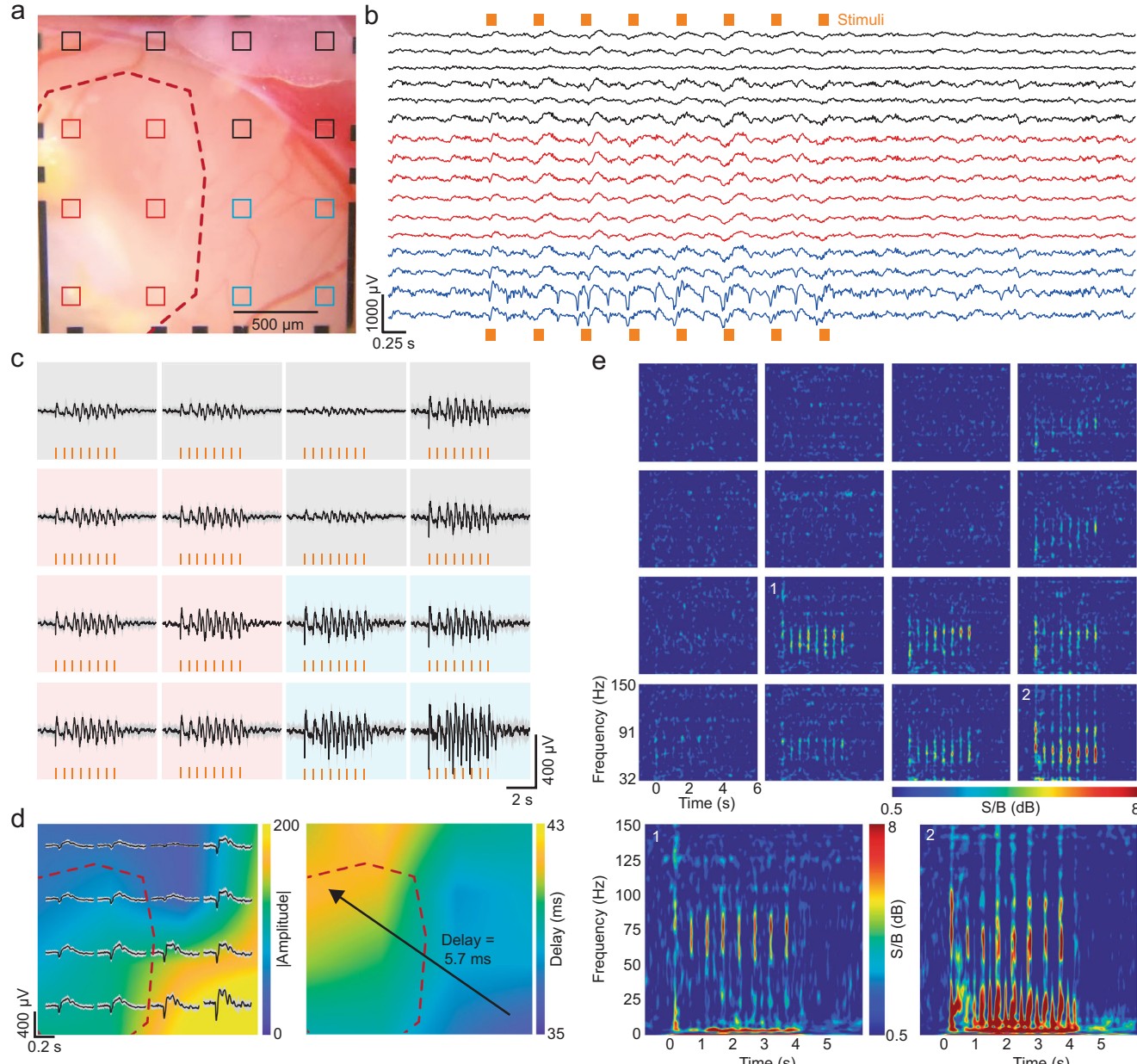

**Fig. 2 | Stimulus-evoked local field potential recordings of organoid and cortex.**
**a** Microphotograph of organoid implantation site (red outline) and surrounding cortex. Electrode pads of the graphene microelectrode array are highlighted; red channels are considered 'organoid channels' whereas blue and black channels overlay host cortex. Blue channels are in the vicinity of the visual cortex. **b** Single trial of a local field potential (LFP) recording during visual stimulation with 100-ms pulses at 2 Hz for 4 s (orange bars) with a white LED to the contralateral eye. All 16 channels are shown; colors of individual traces correspond to channel assignments defined in panel **a**. **c** Trial average of the LFP responses (mean ± sdv of 20 trials). **d** Color map indicating peak amplitude (top) and peak delay (bottom) of the response to the first light pulse as shown in panel **c**. The red dashed lines outline the implantation site, as in panel **a**. The observed delay between cortex and organoid is 5.7 ms (arrow, $p < 0.01$, Student's one-tailed t test). Source data are provided as a Source data file. **e** Spectrogram (32–150 Hz) of the response (average of 20 trials). Inset: Enlarged spectrogram (1–150 Hz) of the channel overlaying organoid (channel 1) and visual cortex (channel 2). The recording shown in panels **b**–**e** was performed 69 dpi; the results shown are representative for a total of five animals.

channel 2). By 3 weeks post-implantation, organoids showed LFP responses matching the amplitude of those in surrounding cortex, suggesting that xenograft neurons established synaptic connections with the surrounding cortex tissue and received functional input from mouse brain.

**Multi-unit activity is modulated in presence of sensory stimuli**
Next, we investigated whether the spiking activity of organoid cells are modulated by the neural activity of the surrounding cortex using recordings with transparent graphene microelectrodes. We evaluated

multi-unit activity (MUA) in 0.5–3 kHz band-pass filtered data. Similar to high-frequency LFPs, high-frequency MUA signals are more strongly attenuated by tissue and therefore primarily reflect local neuronal firing (within 100–200 μm of the recording electrode)[15,34–37]. Figure 3 shows MUA recordings. We chose two representative channels (red, channels O1 and O4) overlaying the organoid xenograft and two channels (blue, channels C2 and C7) overlaying surrounding cortex (Fig. 3a). In all channels, we observed spontaneous MUA events, which were defined as time points where the MUA crossed a pre-defined threshold of −3 to −4 times of its standard deviation (Fig. 3b).

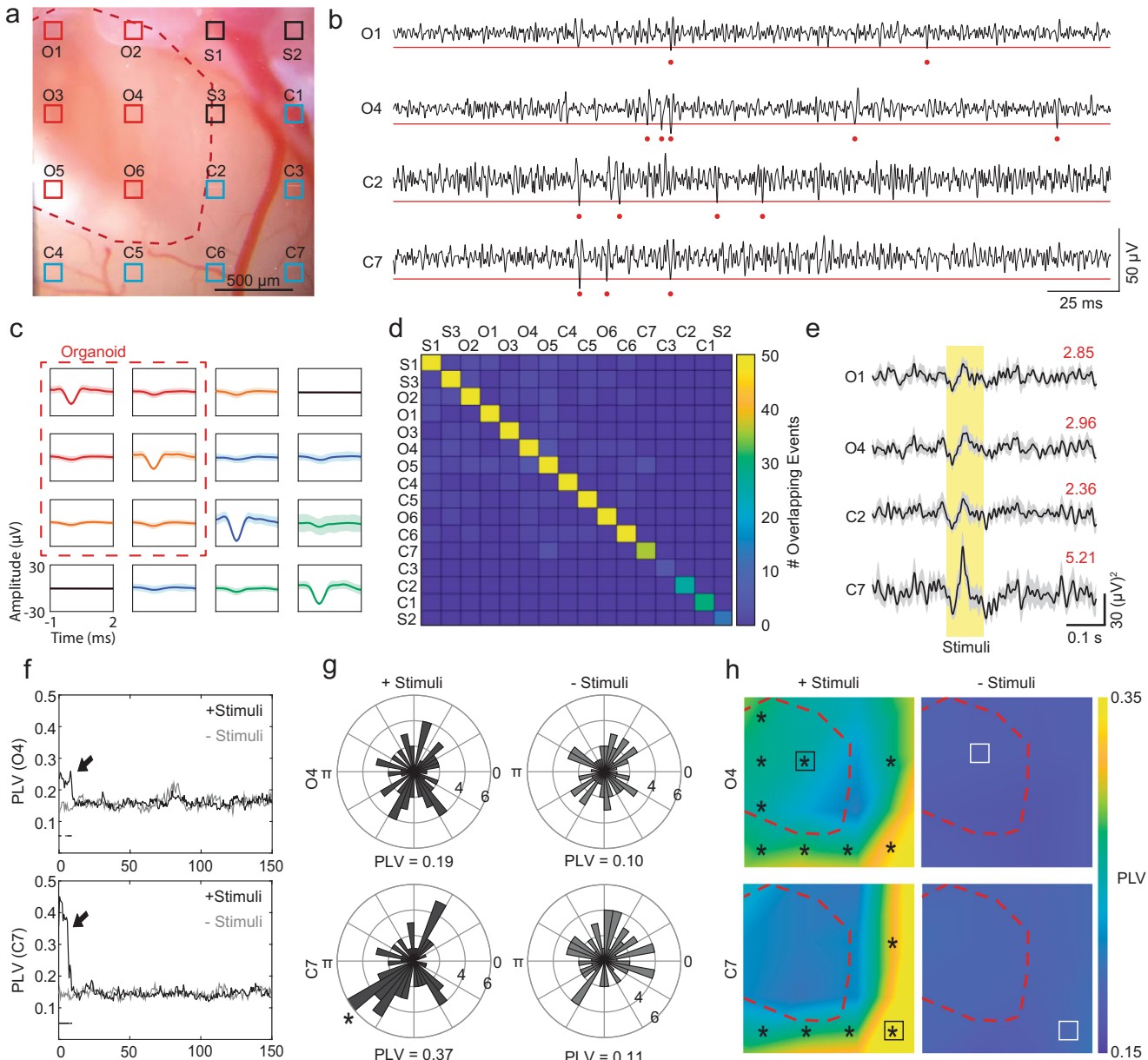

**Fig. 3 | Multi-unit activity of organoid and cortex. a** Microphotograph of implantation site (red outline) and surrounding cortex. Electrode pads of the graphene microelectrode array are highlighted; channels C1-C6 are defined as 'organoid channels' whereas channels O1-C7 are defined as 'cortex channels.' **b** Recording of spontaneous MUA from channels O1, O4, C2, and C7 on 21 dpi. Dots indicate MUA events crossing the −3.5 sdv threshold. **c** Representative examples of event-averaged MUA traces (data presented as mean ± sdv) for four channels (O1, O4, C2, and C7, indicated with different color traces) showing the spatial localization of spontaneous MUA events. **d** Count of overlapping events across channels after binning into 1-ms epochs shows almost no overlapping of events. **e** Trial average (6 trials, data presented as mean ± sdv) of rectified MUA activity in response to the first pulse of a 4-s train of 100-ms light pulses (yellow region) at 2 Hz delivered to the contralateral eye. Red numbers indicate the signal-to-noise ratio of the peak response. **f** Average phase locking value (PLV) vs. frequency for channel O4 (organoid channel, top) and

channel C7 (cortex channel, bottom). Arrows point to frequency regions where the PLV was higher during stimulation (black) compared to periods without stimulation (gray). Black dots indicate frequencies where PLVs are significantly different (95% bootstrap confidence interval). Theta PLV at 4−6 Hz is further analyzed in panels **g** and **h**. **g** Radial histograms of 5-Hz LFP signal phases during MUA events in channels 2 and 4. Asterisk indicates significant phase locking (*p* = 0.0216) using Rayleigh's test for non-uniformity of circular data (one-sided) and is placed at the preferred angle. **h** Color map of the spatial extent of phase locking; stimulus-induced MUA events of both channels show strongest phase locking to theta oscillations closest to the visual cortex (bottom right). Asterisks indicate *p* < 0.05 between episodes with and without stimulation (only shown on the left plot) using Rayleigh's test for non-uniformity of circular data (one-sided). Source data for **d**, **f**, **g**, and **h** and exact *p*-values for **f** and **h** are provided as a Source data file. The results shown are representative for a total of three animals.

Spontaneous MUA events remained relatively stable over the 8- to 11-week experiments, with spiking rates around 2 Hz (*n* = 4 animals, Supplementary Figs. 5 and 6), consistent with activity of similarly aged organoids measured in vitro[4]. To verify that the MUA recordings were localized spatially and unique to each channel, we evaluated event-triggered MUA traces[38]. Event-triggered MUA averages were computed

by taking a target channel's MUA event times and, for all 16 channels, computing the average MUA waveform from 1 ms before to 2 ms after each of the target channel's events. If the same neural events were picked up by multiple channels, then similar MUA averages would appear across several, nearby channels. And if the channels recorded independent neural activity, then the MUA event deflection would only

appear in the target channel and all other channels would average to zero. Our results showed that MUA events recorded from each channel overlaying the organoid or cortex were local and did not show any spread across other channels (Fig. 3c, Supplementary Fig. 7b, c). Counting the number of overlapping events between channels after 1-ms binning, we saw that there was almost no co-occurrence of events (Fig. 3d, Supplementary Fig. 7d). The number of MUA event overlaps between the organoid and cortex channels were not statistically significant and fell within the chance range (Supplementary Fig. 7e), supporting that MUA recordings originate from local neural firing and therefore recordings from electrodes overlaying the organoid are generated locally by the organoid neurons and cannot originate from the surrounding cortex channels.

Increases in MUA power were observed in channels overlaying both the organoid and cortex in response to the onset of visual stimulation (Fig. 3e). Notably, there was also a change in MUA power within organoid channels (Fig. 3e, channels O1 and O4). We then investigated whether MUA in the organoid was modulated by LFP in the organoid and surrounding host cortex by analyzing the phase-locking values (PLV) of the MUA to LFP. PLVs close to zero indicate MUA events at random LFP phases, whereas PLVs equal to 1 indicate MUA events in synchrony with the LFP signal. Buzsaki et al.[27] suggested that "verification of the local nature of the signal always requires the demonstration of a correlation between the LFP and local neuronal firing." Therefore, we performed phase locking analysis to investigate local nature of LFPs. Demonstration of phase locking between LFP and MUA, which we verified to be local to channels, supports that our LFP recordings with transparent electrodes are primarily recording locally generated neural activity. Previous studies have shown that PLVs increase or decrease during light stimulation[20,39–41] and whisking[42] in in vivo experiments with intact cortex. We observed that PLVs increased in the delta and theta bands in both cortex and organoid channels (dots below traces indicate 95% bootstrap confidence interval) (Fig. 3f). The MUA-theta PLV during times with stimulation versus times without stimulation is further visualized by plotting the 5 Hz (center of the 4–6 Hz theta band) LFP phases during each MUA spike on a radial histogram (Fig. 3g, $n = 60$ spikes for channel O4 and $n = 67$ spikes for channel C7) . To determine the spatial extent of MUA phase locking, we compared MUA events in channels O4 and C7 to LFPs in all 16 channels. During visual stimulation, MUA signals from organoid and cortex channels showed significantly increased phase-locking to the theta band of LFP signals in electrode channels close to the channel of interest (where MUAs were measured) and most strongly close to channels in the vicinity of visual cortex (Fig. 3h, asterisks mark 95% bootstrap confidence interval of significant increase in stimulated vs. non-stimulated PLV). The significant change in organoid phase locking resembles that of surrounding cortex and only occurred during stimulation, suggesting that the organoid formed functional synaptic connections with the surrounding mouse cortex. These synaptic connections were later verified using post-mortem immunofluorescence staining for human cytoplasm (STEM121), human nucleoli (NM-95), post-synaptic density (PSD95), and human- (hSyn) and non-species-specific synaptophysin (Syn) markers.

## Transplanted organoid responds differently to anesthesia compared to host cortex

Having seen functional responses to sensory stimuli and modulation of spiking activity in the transplanted organoid in awake recordings, we next asked how anesthesia affects spontaneous neural activity in organoids versus the surrounding host cortex. Like adult-born neurons integrating into pre-existing neuronal networks in hippocampus and olfactory bulb[43,44], we hypothesized that the organoid xenograft initially receives mainly local inputs while long-range projections, e.g., from thalamic nuclei and neuromodulatory centers, are largely absent.

As anesthesia has been shown to affect activities of long-range projections to cortex[45], we hypothesized that xenografted organoids exhibit a different type of activity under anesthesia compared to host cortex. Here, we analyzed the burst suppression patterns characterized by alternating periods of high LFP activity (bursts) and periods of relative silence (suppression)[46] induced by anesthesia with 1.5% isoflurane (see, e.g., Supplementary Fig. 8). In a representative recording performed 3 weeks after implantation, shown in Fig. 4, six channels were marked as overlaying the organoid and eight overlaying the cortex (Fig. 4a). Isoflurane anesthesia induced a larger reduction of neuronal activity in the organoid versus surrounding cortex, which was more severe for the gamma band known to be associated with long-range neuromodulation[47] (Fig. 4b–d and Supplementary Fig. 9). Normalized power ratio exhibits a larger contrast under anesthesia compared to the awake state, particularly more prominent for higher frequencies (Fig. 4b). The weaker activity in the organoid compared to cortex activity could be explained by an absence of cholinergic innervations, which are associated with gamma activity[47]. In the awake state, organoid activity was in general lower than in the surrounding cortex with no selectivity for specific frequency bands (Fig. 4e, f). Thus, the organoid reacts differently to anesthesia versus surrounding cortex. Our results suggest that the organoid cells behave similarly to adult-born neurons which innervate with local neurons but lack long-range projections[48].

## Organoid grafts are vascularized by the host and integrate with surrounding cortex

To examine morphological integration of the cortical organoids with the surrounding host tissue, including vascularization by the host blood vessels, we used in vivo two-photon imaging and post-mortem immunostaining. During every recording session, the macroscopic structure of the implantation site was inspected using brightfield stereomicroscopy unimpeded due to the transparency of the microelectrodes (see, e.g., Fig. 2a). Nine to ten weeks after implantation, mice underwent two-photon microscopy after injection of the intravascular tracer Alexa Fluor 680 Dextran[49]. Organoid implantation regions in all six mice contained mouse vasculature. A representative result from two-photon imaging is shown in Fig. 5a. After acquiring a low-magnification map of the entire exposure (Fig. 5a, left), we acquired high-resolution Z stacks within the implant region (Fig. 5a, center) and surrounding cortex (Fig. 5a, right). Vascularization of the organoid was clearly visible, confirming integration into the host cortical tissue (see Supplementary Video 1 for a 3D projection of the vasculature within the implant region). The vessel density was lower in the organoid region compared to the surrounding cortex, as observed in our previous study[9].

Eight to eleven weeks after implantation, animals were sacrificed, and brains underwent histological analyses with antibodies detecting human cells (NM-95 antibody specific for human nucleoli), endothelial cells (CD31), and neuronal nuclei (NeuN). We found NM-95-positive (i.e., human) cells in all animals and noted some variability in the xenograft size across animals. Figure 5b shows a representative coronal section from one animal after NM-95 staining. The regions of organoid implantation stained clearly for NM-95, confirming that human cells survived in mouse cortex for the entire experiment duration. Some of the NM-95-positive cells migrated away from the implantation site (in 6 mice, Fig. 5b, right). Human cells were detected as far as 4 mm away from the implantation site, migrating along the corpus callosum (Supplementary Fig. 10). In adjacent sections stained for endothelial cells (Fig. 5c), we observed blood vessels traversing the xenograft, which is in line with our in vivo two-photon microscopy results and shows vascularization of the xenograft. NeuN co-staining with hematoxylin revealed that ~48% of the cells in the organoid graft had a neuronal phenotype at the final time of recording (Fig. 5d, e). The neuronal phenotype of organoid cells was further supported by the

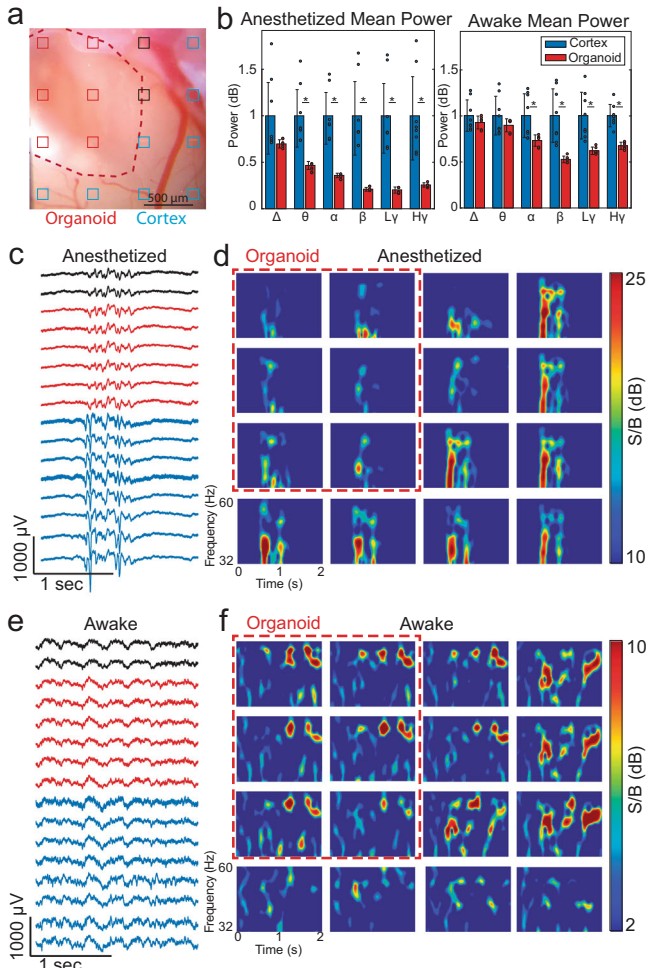

**Fig. 4 | Analysis of cortex and organoid spontaneous LFPs while the animal was awake and anesthetized with 1.5% Isoflurane. a** Microphotograph of implantation site (red outline) and surrounding cortex. Red channels overlay the organoid, blue channels overlay the cortex, and black channels were ambiguous. **b** Average frequency band power while the mouse was anesthetized (left) and awake (right). Frequency powers were normalized to the cortex power (set to 1) and organoid power is shown as a fraction of the cortex power (*$p < 0.05$ in two-sample t-test). Data presented as mean ± sdv (error bars) alongside individual data points from 16 channels in one animal. "Δ" stands for delta, "θ" for theta, "α" for alpha, "β" for beta, "Lγ" for low gamma, and "Hγ" for high gamma. Source data and exact $p$-values are provided as a Source data file. **c** Two-second LFP (low-pass filter, 250 Hz) epoch showing a burst of activity in the anesthesia recording. **d** Spectrogram (low gamma range) during the LFP burst shown in (c). **e** Two-second LFP epoch showing more uniform activity across organoid and cortex. **f** Spectrogram (low gamma range) during the LFP epoch shown in (e). The results shown are representative for a total of five animals.

detection of PSD-95 in the organoid region which is a post-synaptic density marker found solely in excitatory neuron synapses (Fig. 6)[50]. Additional staining for NM-95, proliferating cells (Ki67), and oligodendrocytes (Olig2) yielded the organoid composition as consisting of ~82% human cells, ~5% proliferating cells, and ~7% oligodendrocytes ($n = 3$, Fig. 5e).

Finally, in order to investigate synaptic connectivity between organoid and cortex, we performed immunofluorescence staining for human nucleoli (NM-95), pre-synaptic vesicle protein synaptophysin (Syn), and post-synaptic densities (PSD95). In Fig. 6a–c, we examined the co-localization of Syn (red) and PSD95 (green) with human cells (white) along the boundary (delineated in pink) of the transplanted organoid (Org) and visual cortex (Visual), the boundary (delineated in

pink) of the organoid and retrosplenial cortex (RSC), and within the corpus callosum (CC). We observed clear co-localization of organoid cells with the pre- and post-synaptic markers at the boundary of the mouse visual cortex (Fig. 6a, arrowheads), retrosplenial cortex (Fig. 6b, arrowheads), and corpus callosum (Fig. 6c, arrowheads), suggesting synaptic connectivity. Moreover, we investigated whether the synaptophysin within the organoid was of mouse or human origin and evaluated bi-directional synaptic connections between xenografted organoids and host mouse cortex with post-mortem immunofluorescence analysis. To the best of our knowledge, there is no mouse-specific synaptophysin marker, so we performed triple staining for human cytoplasm (STEM121), Syn, and human-specific synaptophysin (hSyn) ($n = 2$). Focusing on the overlap of Syn and hSyn, we could determine which presynaptic puncta were of human origin. The remaining puncta that labeled positive for Syn but negative for hSyn were counted as presynaptic puncta of mouse origin. Figure 6d, e shows presynaptic puncta and organoid neuronal projections inside the organoid 100 μm from visual cortex boundary and at the organoid center, respectively. We observed mouse presynaptic puncta (hollow arrowheads) colocalized nearby the organoid projections both at the boundary (Fig. 6d) and the center (Fig. 6e) of the organoid, suggesting mouse neurons formed pre-synaptic connections to the organoid. Quantifying the density of human and mouse presynaptic puncta, we observed a significantly greater density of mouse puncta at the boundary compared to center of the organoid, likely due to the proximity to mouse cortex, and a greater density of human puncta at the center of the organoid compared to the boundary, likely due to the dispersion of human cells as they morphologically integrated with mouse cortex (Fig. 6f). We also observed human presynaptic puncta and STEM121 staining in mouse visual cortex near the outer edge of the organoid (Supplementary Fig. 11). Our observations of mouse presynaptic puncta in the organoid and human presynaptic puncta in the mouse cortex support that by the time of the final recording session, organoid and mouse visual cortex had made bi-directional synaptic connections. Mouse presynaptic puncta in the organoid provides evidence for the synaptic connectivity needed by the organoid to generate functional responses to visual stimuli, as observed in electrophysiological recordings.

## Discussion

Recent advances in pluripotent stem cell technology have enabled generation of neuronal cell lines and cortical organoids from hiPSCs isolated from peripheral tissue (e.g., skin biopsies). While these organoids resemble some features of early developmental stages of the human brain, the lack of the natural brain microenvironment in cultured organoids can influence the phenotype and maturation of the reprogrammed neurons. Previously, we have shown that transplantation of human cortical organoids into the adult mouse brain leads to their further differentiation and vascularization[9]. In the present study, we pushed this paradigm further, introducing optically transparent graphene electrode microarrays that enable multimodal longitudinal monitoring of neuronal activity in the graft and the surrounding host neuronal circuits. This setup allowed us to examine morphological integration of the organoid with cortex and revealed organoids functionally integrating with an endogenous sensory cortex over time. With conventional metal electrodes, we would not have had a clear field of view to examine the organoid graft and proximity to sensory cortex without needing to remove the electrodes and disrupt the implantation site. This combination of stem cell and neurorecording technologies opens opportunities for investigation of human neuronal network-level dysfunction underlying developmental brain disease and how organoids may offer benefits as neural prosthetics to restore lost function of different brain regions.

Our results demonstrate that implantation of graphene microelectrodes does not prevent development and vascularization of the

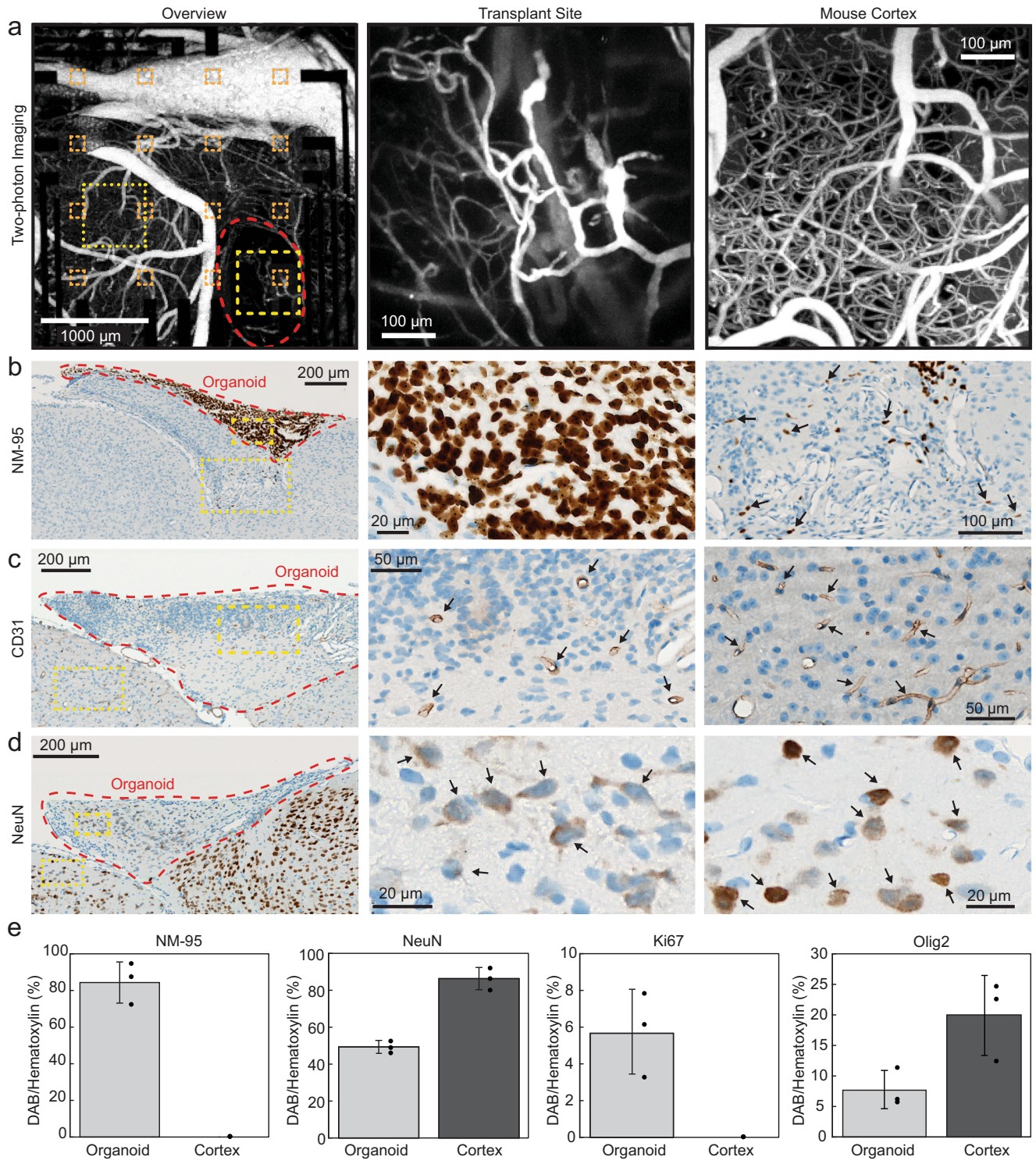

xenograft and allows examination of LFP and MUA neuronal activity along with optical imaging, including two-photon microscopy. With this technology, we demonstrated several differences of organoids' functional responses to sensory stimuli. While LFP signals represent spatial integration of extracellular potentials generated by several mechanisms including synaptic transmission, action potentials, intrinsic currents and ephaptic conduction, its locality and spatial specificity depends on the frequency range. We verified the locality of LFP by (1) correlating the MUA with LFP, (2) inspecting the time delay of LFP signals, (3) analyzing high-frequency LFP, and (4) performing independent component analysis (ICA) as one of our preprocessing

steps. These analyses suggested that LFP and MUA signals recorded by the channels overlapping with the organoid were generated by the activity of the human neurons. The locking of MUA events to LFP of the channels closer to the visual cortex provides further evidence that human neurons participated in the network response to the visual stimulus due to formation of synaptic connectivity with the mouse cortex. The amplitudes of trial-averaged LFPs and spontaneous MUA spikes also increased over time, suggesting improved coupling of organoid with the microelectrode array. Under anesthesia with 1.5% isoflurane, we observed a selective decrease in gamma power in organoid activity versus surrounding cortex. Gamma activity is

**Fig. 5 | In vivo imaging of organoid vascularization and post-mortem immunohistochemical analysis. a** Depth projections (0–650 μm) of image stacks acquired using two-photon microscopy (1240 nm excitation wavelength, 512 × 512 pixel, 3 μm step size) after injection of the intravascular tracer Alexa 680-Dextran. The organoid implant region is outlined in red in the overview image (left) and vasculature of organoid transplant site (center) and mouse cortex (right) are shown. The yellow boxes with thicker outlines highlight representative organoid regions shown in greater detail in the middle panel and the yellow boxes with thin outlines highlight mouse cortical regions shown in greater detail in the right panel. **b** Immunostaining of a 5-μm coronal section with NM-95 antibodies detecting human nucleoli. The implant region (center) contains mainly human cells. Organoid-derived (human) cells (arrows) are also present in surrounding mouse cortex (right). **c** Immunostaining of a 5-μm coronal section with CD31 antibodies detecting endothelial cells. Staining indicates vascularization of the implanted organoid (center, arrows) although at lower density compared to surrounding cortex (right, arrows). **d** Immunostaining with NeuN antibodies detecting neuronal nuclei indicates cells in the implant regions which are neurons. Note that NeuN antibodies show weaker staining of human neuronal nuclei (center, arrows) compared to mouse neuronal nuclei (right, arrows). Sections shown in panels **b–d** were counterstained with hematoxylin (blue); primary antibodies were detected with horseradish-peroxidase-coupled secondary antibodies and diaminobenzidine as chromogenic substrate (brown). **e** Composition analysis of organoid using NeuN, NM-95, Ki67, and Olig2 within organoid and comparison to contralateral cortical tissue. Data are presented as mean values ± sdv (error bars) alongside individual data points, six regions of a slice were analyzed for each bar chart from $n = 2$ mice. Source data are provided as a Source data file. The results shown were repeated and are representative for a total of five (**b**), three (**a**), or two (**c**, **d**) animals.

associated with cholinergic neuromodulation[47]. Assuming lower cholinergic innervation of the xenograft compared to cortex, suppression of cholinergic activity by anesthesia would disproportionally reduce gamma power in the organoid relative to host cortex. Indeed, it has been shown that adult-born neurons gain first local and later long-range connectivity[48]. Anesthetics differ greatly in their mechanisms of action and influence on cortical activity[51,52] and investigating the influence of other anesthetics on xenografted organoids would be an interesting future application of this methodology.

The success of our experiments depended on the engineering of flexible, transparent graphene devices. This setup allowed us to retain chronically implanted microelectrode arrays in the mouse cortices for electrophysiological recording with the ability to image the organoid implantation site at any time. To this end, we designed a light-weight headpost assembly with a protective enclosure for connecting the graphene arrays to the data acquisition system via a ZIF connector during the recording sessions. Along with fusion of the array with the glass window insert, this assembly offered mechanical stability and durability in chronically implanted mice. The present study was designed to obtain histological validation of the xenograft after 11 weeks of in vivo recordings and imaging; longer experiments will be conducted in the future. Another future direction for this technology could be to take advantage of the electrode transparency by incorporating calcium imaging to visualize spiking activity in organoid neurons or rabies viral retrograde tracing of axonal projections between organoid and mouse cortex as others have demonstrated with iPSCs in vivo[53,54] and spheroids in vitro[55,56]. These methods were not established for our specific cell lines at the time of this experiment but could be developed for future use.

While transplantation of human cortical organoids in the mouse brain is still in its infancy, our study takes a step toward comprehensive functional assessment of this biological model system. We envision that, further along the road, this combination of stem cell and neurorecording technologies will be used for modeling disease under physiological conditions at a level of neuronal circuits, examination of candidate treatments on patient-specific genetic background, and evaluation of organoids' potential to restore specific lost, degenerated, or damaged brain regions upon integration.

## Methods
### Animal care
All animal experiments described in this study were conducted in accordance with the National Institutes of Health's Guide for the Care and Use of Laboratory Animals and were approved by the Institutional Animal Care and Use Committee (IACUC) at the University of California San Diego (Protocol S14275). Immune-deficient non-obese diabetic (NOD)/severe combined immunodeficient (SCID) mice, aged 6–8 weeks, were purchased from Jackson Laboratories (JAX Stock: 001303). In this study, female mice with 8–12 weeks of age were used. Animals were kept in autoclaved cages under standard conditions

(20–22 °C, 40–60% relative humidity) on a 12 h light/dark cycle with ad libitum access to food and water.

### Cortical organoid generation
All experiments were approved and performed following the Institutional Review Boards (IRB) and Embryonic Stem Cell Research Oversight (ESCRO) guidelines and regulations. hiPSCs (WT83) used to generate cortical organoids were reprogrammed from skin biopsy-derived fibroblasts[4]. Fibroblasts were donated from neurotypical individuals after informed consent was appropriately given under protocols approved by the University of California, San Diego Institutional Review Board (#141223ZF). hiPSC colonies cultured on Matrigel-coated dishes (BD-Biosciences, San Jose, CA, USA) and fed daily with mTeSR1 (StemCell Technologies, Vancouver, Canada) for ~7 days were dissociated with 1:1 Accutase (Life Technologies, Carlsbad, CA, USA):PBS. hiPSCs were plated into a six-well plate (~4 × 10⁶ cells/well) in mTeSR1 supplemented with 10 μM SB431542 (Stemgent, Cambridge, MA, USA), 1 μM Dorsomorphin (R&D Systems, Minneapolis, MN, USA), and 5 μM Y-27632 (EMD-Millipore, Burlington, MA, USA) and cultured thereafter in shaker suspension (95 rpm at 37 °C). Formed spheres were fed mTeSR1 (with 10 μM SB431542 and 1 μM Dorsomorphin) for 3 days followed by Media1 [Neurobasal (Life Technologies), 1x Glutamax (Life Technologies), 2% Gem21-NeuroPlex (Gemini Bio-Products, Sacramento, CA, USA), 1% N2-NeuroPlex (Gemini Bio-Products), 1% non-essential amino acids (NEAA; Life Technologies), 1% Penicillin/ Streptomycin (P/S; Life Technologies), 10 μM SB431542, and 1 μM Dorsomorphin] for 6 days, every other day; Media2 (Neurobasal, 1x Glutamax, 2% Gem21, 1% NEAA, and 1% P/S) with 20 ng/mL FGF-2 (Life Technologies) for 7 days, daily; Media2 with 20 ng/mL each of FGF-2 and EGF (PeproTech, Rocky Hill, NJ, USA) for 6 days, every other day; and Media2 with 10 ng/mL each of BDNF, GDNF, and NT-3 (all Pepro-Tech), 200 μM L-ascorbic acid (Sigma-Aldrich, St. Louis, MO, USA), and 1 mM dibutyryl-cAMP (Sigma-Aldrich) for 6 days, every other day. Cortical organoids were subsequently maintained in Media2 without supplementation until use. Two-month-old cortical organoids were used for the transplant.

### Surgery for implantation of organoids and graphene microelectrode arrays
Headpost implantation, craniotomy, organoid implantation, and graphene microelectrode array implantation were performed in a single surgery. Twelve to twenty-four hours before surgery, animals received 0.53 mg/mL Sulfamethoxazole, 0.11 mg/mL Trimethoprim, and 40 mg/ mL Ibuprofen through their drinking water. Four hours before surgery, animals received a single injection of 4.8 mg/kg Dexamethasone (i.p.). For anesthesia, animals received a cocktail of 50 mg/kg Ketamine and 5 mg/kg Xylazine that was supplemented with 0.1–1% isoflurane in oxygen for the duration of the surgery. After anesthesia induction, animals received a single dose of 0.05 mg/kg Buprenorphine (s.c.) and 500 mg/kg Cefazolin (i.p.). After hair removal, the scalp was disinfected

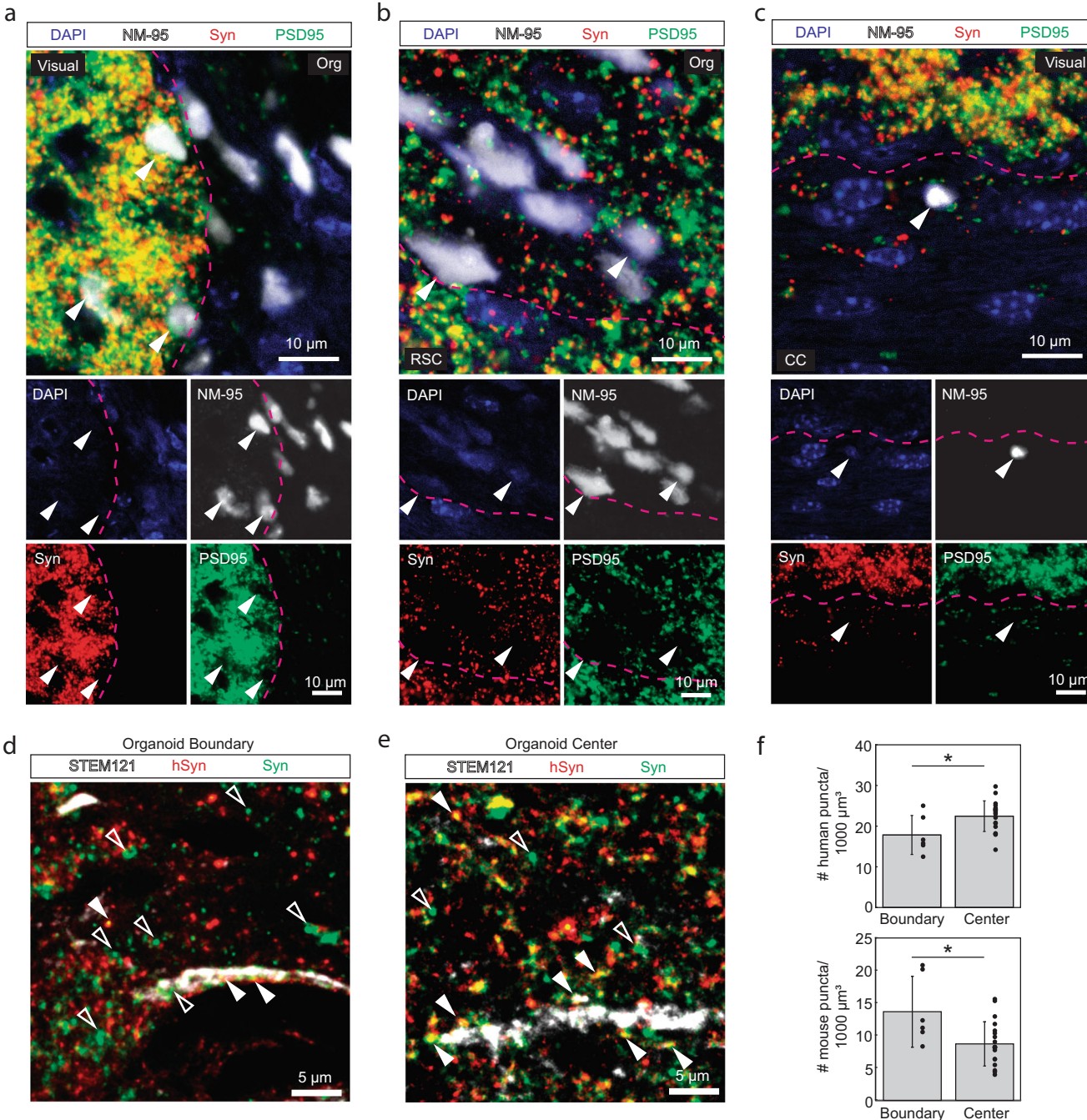

**Fig. 6 | Confocal microscopy of immunofluorescence staining for antibodies against human nucleoli (NM-95), human cytoplasm (STEM121), Synaptophysin (Syn; pre-synaptic terminal vesicle protein), and post-synaptic density (PSD-95) in a region with organoid implant taken from a mouse directly after the last electrophysiology recording at 11 weeks post-implantation. DAPI (blue) was used as counterstain for cell nuclei. a** Co-localization of Syn (red) and PSD-95 (green) with human cells along the boundary (delineated in pink) of the organoid implant (Org) and visual cortex (Visual, arrowheads). **b** Human cells were observed surrounded by Syn, PSD-95, and mouse cells (arrowheads) in a region at the organoid implant and retrosplenial cortex (RSC) boundary (delineated in pink). **c** Human cells (arrowheads) were observed surrounded by PSD-95 and Syn, traveling along the corpus callosum (CC). The boundary between CC and visual cortex is delineated in pink. Note the smaller density of Syn and PSD-95, which is characteristic of the corpus callosum due to longer myelinated axonal projections and less synaptic terminals. **d** Puncta positive for Syn (green) and negative for hSyn (red) were defined as mouse pre-synaptic terminals. The number of mouse pre-synaptic puncta was high at the boundary of the organoid implant (**d**, hollow arrowheads) but still existed at the center of the implant (**e**, hollow arrowheads). Solid arrowheads in (**d**, **e**) indicate puncta positive for both hSyn and Syn, representing human pre-synaptic terminals. **f** The density of human pre-synaptic terminals significantly increased towards the center of the organoid (top bar chart, $p = 0.0176$) and the density of mouse pre-synaptic terminals (i.e., +Syn/-hSyn puncta) significantly decreased towards the center of the organoid (bottom bar chart, $p = 0.008$) (*$p < 0.05$ in two-sample t-test). Data are presented as mean values ± sdv (error bars) alongside individual data points, 11,979 hSyn+ and Syn+ puncta were analyzed from 26 brain regions in one mouse. Source data are provided as a Source data file. The results shown are representative for a total of two animals.

with betadine and isopropanol. Fifty μL of 2% Lidocaine were infiltrated into the skin overlaying the skull. After skin removal, the bone was cleaned, and the skin was attached to the bone at the wound borders using cyanoacrylate glue (VetBond). Then, the skull was etched for 60 s with 35% phosphoric acid gel (Kerr Dental), which was removed by washing with sterile 0.9% NaCl solution. Then, a thin layer of bonding agent (OptiBond, Kerr Dental) was applied to the bone and cured with UV light. The headpost was attached with dental adhesive (Tetric Evoflow). For electrophysiology recordings, a reference electrode (#000 micro-screw) was implanted over the olfactory bulb and secured with dental adhesive. Then, a 3.5-mm diameter round piece of bone overlaying the implant region was removed with a dental drill. After dura mater removal, a ca. 1-mm diameter large piece of retrosplenial cortex overlaying the superior colliculus (−3.5 mm relative to Bregma, +0.75 mm relative to midline) was removed by aspiration. After stopping any bleeding from the aspiration site, a single organoid was placed inside the void. A graphene microelectrode array with 16 channels was bonded to a glass plug consisting of two 3-mm and one 5-mm coverslip glass and placed on top of the craniotomy, which was then sealed with dental adhesive (Tetric Evoflow). A holding cap[57] attached to the headpost protected exposure and microelectrode array wires while animals were in their home cage; it was removed during experiments. At the end of the surgery, animals received a single injection with 100 μL 5% Dextrose in 0.9% NaCl. Animals received 0.05 mg/kg Buprenorphine (s.c.) for 3 days after surgery, and 0.53 mg/mL Sulfamethoxazole, 0.11 mg/mL Trimethoprim, and 40 mg/mL Ibuprofen in the drinking water for 5 days after surgery. Treatment with 0.53 mg/mL Sulfamethoxazole and 0.11 mg/mL Trimethoprim in drinking water was continued for 3–4 weeks after surgery.

## Graphene microelectrode array fabrication

Following our previous fabrication protocol[11,58], 50-μm-thick polydimethylsiloxane (PDMS) was spin-coated onto a 4-inch silicon wafer and annealed at 150 °C for 10 min. The PDMS-coated wafer was used as an adhesive layer to keep the substrate flat during subsequent fabrication steps. Fifty-μm-thick PET film was placed onto the PDMS as an electrode array substrate. Ten nanometers of Chromium and 100 nm of gold were sputtered onto the PET using the Denton Discovery 18 Sputtering System. Metal wires were patterned using photolithography and wet etching (Gold Etchant TFA, Chromium Etchant 1020AC). Monolayer graphene was transferred onto the wafer using the bubble transfer method[59]. The device was dried overnight at room temperature and then baked at 125 °C for 5 min to anneal PMMA substrate wrinkles and increase bonding between the graphene and PET substrate. To remove PMMA, the wafer was submerged in acetone with gentle pipetting for 20 min, then submerged in alternating isopropyl alcohol and deionized water for 10 min in 30-s cycles. Graphene contact pads were patterned using PMGI/AZ1512 bilayer lithography and oxygen plasma etching (Plasma Etch PE100). To remove organic residue from the graphene surface, a four-step cleaning method was used: (1) soaking the wafer in AZ 1-Methyl-2-pyrrolidon for 5 min, (2) soaking the wafer in Remover PG for 5 min, (3) soaking the wafer in acetone for 10 min, and (4) rinsing the wafer with alternating deionized water and isopropyl alcohol for 10 min in 30-s increments. Steps 1 and 2 removed AZ1512 and PMGI, respectively. Steps 3 and 4 removed any remaining organic residue from the wafer surface. The arrays were then encapsulated with 7-μm-thick SU-8 2005 that was spin-coated onto the wafer and exposed under UV light to pattern active electrode area openings. A final clean of the wafers was done using 10 min of alternating deionized water and isopropyl alcohol. Then, the wafers were baked for 20 min starting at 125 °C and gradually increasing the temperature to 135 °C to seal the SU-8 encapsulation layer. The PET substrate was peeled from the PDMS-coated wafer

and the arrays were cut out for electrochemical characterization and implantation in mice.

## Electrochemical characterization of graphene microelectrode arrays

Electrochemical impedance spectroscopy (EIS) was performed with Gamry Reference 600 potentiostat in 0.01 M phosphate-buffered saline (Sigma-Aldrich P3813 dry powder dissolved in deionized water). A three-electrode configuration was used with Ag/AgCl as the reference electrode and platinum as the counter electrode. EIS were measured from 100 kHz to 1 Hz at open circuit potential. The entire electrode configuration system was placed in a self-made Faraday cage to eliminate noise.

## Electrophysiology data recording

Animals were anesthetized with 5% Isoflurane (induction) and kept at 1.5% for maintenance. While under anesthesia, they underwent head fixation, the protective cap was removed, and the array connected to the recording setup. After recording for 5–10 min while the animal was anesthetized, anesthesia was removed, and the animal allowed to recover. After ca. 10 min, recording was continued.

Electrophysiological recording was conducted with the RHD2000 amplifier board and RHD2000 evaluation system (Intan Technologies). The sampling rate was set to 20 kHz and DC offset was removed with the recording system's built-in filtering above 0.1 Hz. Intan data was imported into MATLAB (MathWorks) and analyzed using custom scripts.

## Visual light stimulation

A white-light LED was connected to a fiber optic cable to deliver light stimulation. The light source was placed ~1 m away from the animal and electrophysiology recording equipment to avoid crosstalk and electrical noise. The tip of the fiber-optical cable was placed ~3 mm from the contralateral eye of the animal to illuminate the entire eye. Five- to 100-ms light pulses (i.e., photic stimulation) were delivered at frequencies of 1, 2, 5, 10, 15, 55, and 85 Hz for intervals of 1–5 s with 10–20 repetitions per experiment. Stimulation was controlled through a DAQ system (National Instruments) driven by custom-written MATLAB codes. To synchronize stimulation and recording, a stimulus-locked trigger signal was delivered from the DAQ system to the Intan recording system.

## Video recording

During experiments, video recordings of the animal were performed to detect body and whisker motion. A white-light LED was placed in the field of view of the camera that was used to synchronize video recordings with trial trigger from electrophysiology recordings and stimulation.

## Two-photon imaging

Animals were anesthetized with 3–5% Isoflurane in oxygen and received an intravenous injection of 100 μL Alexa 680-Dextran. Animals were placed inside the recording platform on a heating blanket. Anesthesia was continued with 1.5% Isoflurane in oxygen. Images were obtained using an Ultima two-photon laser scanning microscopy system from Bruker Fluorescence Microscopy equipped with an Ultra II femtosecond Ti:Sapphire laser (Coherent) coupled to an Optical Parametric Oscillator (Chameleon Compact OPO, Coherent) tuned to 1240 nm. Alexa Fluor 680 was imaged using a GaAsP detector (H7422P-40, Hamamatsu). We used a ×4 objective (XLFluor4x/340, NA = 0.28, Olympus) to obtain low-resolution images of the exposure. A ×20 water-immersion objective (XLUMPlanFLNXW, NA = 1.0, Olympus) was used for high-resolution imaging. The microscope was operated using PrairieView software (Bruker).

## Electrophysiology data analysis and statistics

Data was analyzed in MATLAB (Mathworks, v2019b). Figure preparation was performed with Illustrator (Adobe, version 25.2.3). Data was pre-processed to remove common artifacts from awake mouse motion, 60 Hz power line noise, and shared volume conducted signals using Independent Component Analysis with the jadeR algorithm in EEGLab[16,17,60,61]. Electrodes with impedances above 4 MΩ were excluded from analysis. To extract local field potentials (LFP), raw electro-physiological recordings were low-pass zero-phase filtered below 250 Hz using an 8th order Chebyshev filter (designfilt.m, filtfilt.m). To extract multi-unit activity (MUA), raw data was band-pass filtered between 0.5 and 3 kHz using a 6th order Chebyshev filter (designfilt.m, filtfilt.m). MUA power was calculated by full-wave rectifying and then low-pass (<100 Hz) filtering the MUA. MUA power averages were taken following stimuli and peak signal-to-noise ratio (SNR) was calculated as $10*\log_{10}(peak/baseline)$ with baseline as the average power 1 s before stimuli onset. For spectral analysis, spectrograms using Morlet wavelet spectral analysis were computed of the entire recordings (10 or 20 trials) and trial averages within each recording following light stimulation were computed (Supplementary Fig. 12). The peaks of each LFP onset were determined for each electrode channel (findpeaks.m), and color maps showing the amplitude and offset were created for a visual representation of signal propagation across the brain and electrode array. Statistical significance for delay times was computed by shuffling the 16 channel delay times 1000 times, computing the average and standard deviation of delay times between a channel pairing, then using Student's one sample, two-sided t test (ttest.m) to compute the p-value. For MUA analysis, MUA events were detected as points at which the MUA signal crossed −3 to −4 times the standard deviation threshold (depending on recording SNR). Event-triggered MUA averages were computed by taking a target channel's MUA event times and, for all 16 channels, computing the average MUA waveform from 1 ms before to 2 ms after each of the target channel's events. MUA event overlap was determined by binning the MUA events into 1-ms bins then counting the number of overlapping events across channels. Significance was calculated by randomly shifting the event train of one channel 10,000 times and counting the number of overlaps against another channel, with p-value as the number of times the shifted overlap count exceeded the non-shifted case count. Phases of the LFP frequency spectral component at each timepoint and frequency from 1 to 250 Hz were determined using the multitaper method (modified function mtspecgramc.m, Chronux toolbox[62]) (with a time-bandwidth product of 3–5, 5–9 tapers, zeros padding, and a 1-s window size), and the phase locking value (PLV) was calculated as the absolute value of the circular average of individual phases for each MUA event timepoint (1000–2000 bootstrapped samples of size 50 events sampled from each trial containing 60–100 spikes). PLVs with 95% bootstrap confidence interval were considered significant. Then, the multi-taper phase for each frequency of LFP at the MUA events was compared within the same channel and compared across channels. Phases were averaged and plotted as polar histograms (within channel) and as color plots (across channels) showing the relationship between MUA of each channel and the phase of all other channels. Rayleigh's test for non-uniformity was conducted using Circular Statistics Toolbox for MATLAB[63]. PLV analysis was done for both LED stimuli on and off intervals to compare the PLV between stimulation parameters. For anesthesia analysis, the average power at each frequency band (computed using Morlet wavelet spectral analysis) was computed for each electrode channel. Then, the average powers of organoid and cortex channels for each frequency band were computed and compared using Student's two-sample, one-sided t-test (ttest2.m).

## Histology and immunohistochemistry

Eight to eleven weeks post implantation, mice were sacrificed and transcardially perfused (Heparin-PBS, then 4% PFA in PBS with 2% Sucrose). Brains were removed and fixed in 4% PFA in PBS with 2% sucrose for 12–18 h. After fixation, the brains were placed into 20% sucrose for about 2 weeks and then transferred into 0.01 M phosphate-buffered saline until processed for immunohistochemistry (IHC). Brains were dehydrated and embedded in paraffin. Sixty to hundred coronal slices spanning the region of organoid implantation were cut using a microtome at 5 μm thickness. The slices were then stained with antibodies for NM95/human nucleoli (1:300, Abcam ab190710), NeuN (1:300; EMD Millipore), CD31 (1:300, Dianova), STEM121 (Takara, Y40410; 1:75), hSyn (Invitrogen 14-6525-80; 1:750), or total synapto-physin (Invitrogen MA1-213; 1:3000). Slides were stained on a Ventana Discovery Ultra (Ventana Medical Systems). Antigen retrieval was performed using CC1 (Tris-EDTA based buffer with pH 8.6, Ventana Medical Systems) for 24–40 min at 95 °C. Primary antibodies were incubated on the sections for 32 min at 37 °C. To minimize mouse-on-mouse non-specific staining issues, sections were incubated with a rabbit anti-Ms (IgG1, IgG2a, IgG3; Abcam ab133469; 1:1000) and then detected with a tertiary HRP polymer-linked anti-Rb (OmniMap; 05266548001; Ventana Medical systems). For immunohistochemistry, antibody presence was visualized used diaminobenzidine as a chromogen followed by hematoxylin as counterstain. Slides were rinsed, dehydrated through alcohol and xylene and sealed with a coverslip. A parallel set of sections was stained for H&E. For multi-channel IF, sections were stained sequentially with an antibody stripping step between each antibody/fluorochrome pair using the TSA-Alexa fluor dyes (Alexa 488, 594, 647). Slides were scanned with a slide scanner (Axio Scan.Z1, Carl Zeiss AG). Confocal images were acquired using a Leica SP8 confocal microscope at the University of California at San Diego Neurosciences Microscopy Core. Immunofluorescence images were analyzed using Leica Microsystems LAS AF Lite (Version 2.6.0 build 7266). Immunohistochemistry images were analyzed using ZEN 3.2 (blue edition, Carl Zeiss Microscopy GmbH). Individual immuno-fluorescence channels of merged images were brightness adjusted for better visualization (linear LUT with approximately 10 minimum and 150 maximum). Immunohistochemistry images were adjusted to set gamma approximately equal to 1 for better visualization.

## Reporting summary

Further information on research design is available in the Nature Portfolio Reporting Summary linked to this article.

## Data availability

Source data with quantifications for tables, plots, charts, and statistics are provided with this paper in the Source data file. The raw data that support the findings of this study are available on request from the corresponding authors. Source data are provided with this paper.

## Code availability

The code for preprocessing and processing neural recordings and generating figures are available at the following Github repository: M. N. Wilson, M. Thunemann, X. Liu, Y. Lu, F. Puppo, J. W. Adams, J. H. Kim, M. Ramezani, D. P. Pizzo, S. Djurovic, O. A. Andreassen, A. A. Mansour, F. H. Gage, A. R. Muotri, A. Devor, and D. Kuzum. 'Multimodal monitoring of human cortical organoids implanted in mice reveal functional connection with visual cortex', mwilsonUCSD/multimodal_organoids, 10.5281/zenodo.7375877, 2022. https://github.com/mwilsonUCSD/multimodal_organoids[64].

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

## Acknowledgements

This project is funded through National Institutes of Health (NIH) BRAIN initiative (R21EY030727 to A.D. and D.K., R01MH111359 and R01DA050159 to A.D., and DP2 EB030992 to D.K.), Research Council of Norway (223273, 248828, and 283798 to O.A.A. and S.D.), K.G. Jebsen Stiftelsen (to O.A.A. and S.D.), South-Eastern Norway Regional Health Authority (#2022087 to S.D.), NIH (R21 EY029466 and R21 EB026180 to D.K. and R01MH108528, R01MH109885, and R01MH1000175 to A.R.M.), National Science Foundation (NSF) (ECCS-1752241, and ECCS-2024776 to D.K.), Office of Naval Research (ONR) (N000142012405 and N00014162531 to D.K.), the JPB Foundation (to F.H.G.) and the AHA-Allen Initiative award (19PABH134610000 to F.H.G.). The fabrication of the microelectrodes was performed at the San Diego Nanotechnology Infrastructure (SDNI) of UCSD, a member of the National Nanotechnology Coordinated Infrastructure, which is supported by the National Science Foundation (Grant ECCS-1542148). The acquisition of confocal images was conducted at the University of California at San Diego Neurosciences Microscopy Core, which is supported by the National Institute of Neurological Disorders and Stroke (Grant NINDS P30NS047101). We thank Mary Lynn Gage for her editorial assistance and Qun Cheng for contributions to surgical implantation of organoids and microelectrode arrays.

## Author contributions

M.N.W., M.T., A.D., and D.K. were responsible for study conception and design. M.N.W. and M.T. performed experiments. M.N.W., M.T., X.L., M.R., D.K., and A.D. contributed to data analysis and interpretation. M.N.W., Y.L., and J.K. contributed to microelectrode array fabrication and testing. F.P. and J.W.A. contributed to organoid generation. M.T. and A.A.M. contributed to surgical implantation of organoids and microelectrode arrays. D.P.P. conducted and analyzed immunohistochemistry. S.D., O.A.A., F.H.G., and A.R.M. contributed to data interpretation. M.N.W. and M.T. prepared figures and drafted the manuscript. M.N.W., M.T., F.P., J.W.A., D.P.P., A.A.M., F.H.G., A.R.M., D.K., and A.D. revised the manuscript. All authors approved the final version of the manuscript.

## Competing interests

The authors declare the following competing interests: A.R.M. is a co-founder and has equity interest in TISMOO, a company dedicated to genetic analysis and brain organoid modeling focusing on therapeutic applications customized for autism spectrum disorder and other neurological disorders with genetic origins. The terms of this arrangement have been reviewed and approved by the University of California San Diego in accordance with its conflict of interest policies. The remaining authors declare no competing interests.
