## [Peer Review File · Nature Communications]

Multimodal monitoring of human cortical organoids implanted in mice reveal functional connection with visual cortexREVIEWER COMMENTS

Reviewer #1 (Remarks to the Author):

In this paper, Wilson and colleague conducted an experiment based on transparent graphene microelectrode arrays and two-photon imaging for longitudinal, multimodal monitoring of human organoids transplanted into the mouse cortex. The transparency of graphene microelectrodes allows visual and optical in vivo inspection of the transplanted organoid and the surrounding cortex throughout the chronic experiments where local field potentials and multi-unit activity (MUA) are recorded during spontaneous activity and visual stimuli. It is nice work to develop platform than can utilize the human organoid for Neuroscience as well as translational research. However, I do have some concerns about the results shown and I hope the authors provide answers to these questions.

Major comments :

1. LFP recording from organoid while visual stimuli were given

One of my main concerns is the LFP recording of organoids, as the authors claim. As the authors are well aware, LFP is "the spatial summation of membrane potentials generated from multiple sources". LFP recordings are not so specific when it comes to determining the source of the signal being generated. The authors stated that they analyzed the latency of LFP signals from the organoids and that they were comparable to the results of previous electrophysiological recordings. However, these results still did not provide direct evidence that the LFP recordings actually came from the transplanted organoids. If the authors had performed calcium imaging of the organoids, it would be much easier to identify the signal sources that are organoids.

2. Visualization of the organoid with 2p imaging

The authors used in vivo real-time two-photon imaging to visualize the implanted organoids. I wondered why the authors did not use fluorescent markers to track transplanted organoids, as in the study by Rios and Clevers (Nature Methods, 2018). It would be much easier to trace the evolution of organoids with a fluorescence marker and will provide direct evidence for morphological fusion between the organoid and mouse cortex..

Minor

1. Did you try different anesthesia, other than isoflourane?

2. Have you tried any anesthesia other than isoflourane? MoA is different for each anesthetic, so if different anesthetics produce different consequences, it may provide useful insight into organoid transplantation research.

3. In Supplementary Figure 3, the change in MUA spike rate over time is markedly different in 4 mice. In particular, the data from Mouse 3 stands out. What's the reason?

Reviewer #2 (Remarks to the Author):

The paper by Wilson et al., describes new method combining transplantation of cortical organoids derived from human iPSC and transparent graphene microelectrodes. The study shows that grafted organoids reveal that peripheral stimulation evokes electrophysiological responses in the organoids. Although, this studies technically very sound and advances our capacity to monitor activity of transplanted organoids it has serious shortcomings. Firstly, it is very poor in morphological representation of the transplanted organoids and do not show even neurons, only single stained human cells with unknown phenotype. Authors need to demonstrate what is the neuronal survival and composition of the organoids at a time of recording and also cytoarchitecture of the grafted and surrounding tissue. Secondly, authors fail to demonstrate which brain structures (sub-cortical and cortical) make synaptic connections with transplanted organoid neurons and whether those grafted neurons establish synaptic connections of the host brain. These issues have been successfully addressed in recent studies using rabies virus-based retrograde labeling in combination with other methods to demonstrate functional afferent and efferent connection is the cortical neurons derived from human iPSCs and transplanted in lesioned cortex. Unfortunately, authors do not even mention

these papers. I believe when authors discuss functional connectivity and integration of grafted organoids it is of great importance to demonstrate their connectivity to the host brain in order to fully understand the usefulness of present approach for modeling and exploring human brain development and dysfunction.

REVIEWER COMMENTS

Reviewer #1

We would like to thank the reviewer for kindly reviewing our paper and highly appreciate their insightful comments. We have revised the manuscript according to the suggestions and believe that the revisions have improved the paper. Please find below our responses (in regular fonts) to each specific comment (in italic fonts) provided by the reviewer.

In this paper, Wilson and colleague conducted an experiment based on transparent graphene microelectrode arrays and two-photon imaging for longitudinal, multimodal monitoring of human organoids transplanted into the mouse cortex. The transparency of graphene microelectrodes allows visual and optical in vivo inspection of the transplanted organoid and the surrounding cortex throughout the chronic experiments where local field potentials and multi-unit activity (MUA) are recorded during spontaneous activity and visual stimuli. It is nice work to develop platform than can utilize the human organoid for Neuroscience as well as translational research. However, I do have some concerns about the results shown and I hope the authors provide answers to these questions.

Major comments:

1. LFP recording from organoid while visual stimuli were given

One of my main concerns is the LFP recording of organoids, as the authors claim. As the authors are well aware, LFP is "the spatial summation of membrane potentials generated from multiple sources". LFP recordings are not so specific when it comes to determining the source of the signal being generated. The authors stated that they analyzed the latency of LFP signals from the organoids and that they were comparable to the results of previous electrophysiological recordings. However, these results still did not provide direct evidence that the LFP recordings actually came from the transplanted organoids. If the authors had performed calcium imaging of the organoids, it would be much easier to identify the signal sources that are organoids.

Our response: We thank Reviewer 1 for recognizing our technology for beneficial use with organoid and translational research. Regarding Reviewer 1's first comment, as a general statement it is true that LFPs are the spatial summation of membrane potentials generated from multiple sources. However, its locality and spatial specificity depends on the frequency range. That is why we did not limit our analysis to broadband LFP but focused on high gamma and MUA bands as well. LFP in the high gamma frequency range (>100 Hz) has been suggested to be strongly correlated with local spiking activity (Scheffer-Teixeira et al. 2013; Einevoll et al. 2013; Petterson and Einevoll 2008; Rasch et al. 2008; Ray et al. 2008; Buzsaki, Anastassiou, and Koch 2012). MUA has been widely accepted as a reflection of local spiking activity by the neuroscience community (Bastos and Schoffelen 2016; Buzsaki, G., C. A. Anastassiou, and C. Koch. 2012; Henrie and Shapley 2005; Super and Roelfsema 2005; Kajikawa and Schroeder 2011; Chestek et al. 2020). In this work our goal was to record functional electrophysiological responses from organoids implanted in vivo. Therefore, we did not perform calcium imaging and instead we preferred to directly record MUA and LFPs and performed four additional analyses described in the next paragraph to prove that the signal source is indeed the organoid. Expression of calcium indicators would require establishing a protocol to transduce or genetically modify the organoids and may take an additional year or more of study on that topic alone when it was not our focus for this paper. Since it was already demonstrated by our coauthors Dr. Mansour and Dr. Gage

(Mansour et al. 2018), it would not add anything in terms of novelty and new findings to our paper. Furthermore, we have previously shown that surface potentials recorded with transparent graphene electrodes strongly correlates with cellular spiking (Lu et al. 2018; Thunemann et al. 2018).

To identify the signal sources of electrically recorded neural activity in this manuscript, we 1) performed multiunit activity (MUA) (0.5-3 kHz) recordings and showed that they are locally generated and correlated the MUA with LFP through phase locking analysis, 2) analyzed high-gamma LFP, 3) performed independent component analysis (ICA) as one of our preprocessing steps, and 4) inspected the time delay of LFP signals. We have provided further details on how these four methods prove that our electrical recordings were local to organoid or cortical tissue and incorporated further clarification into the manuscript as follows.

1. MUA Locality and Correlation with LFP

In their seminal paper “The origin of extracellular fields and currents — EEG, ECoG, LFP and spikes” (Buzsaki, Anastassiou, and Koch 2012), Koch et al. states that “verification of the local nature of the signal always requires the demonstration of a correlation between the LFP and local neuronal firing.” Following this method, we first focused on proving the local generation of MUA signals by the organoid and then extended it to studies of correlation between MUA and LFP through phase locking analysis.

MUA has been widely accepted as a reflection of local spiking activity by the neuroscience community (Bastos and Schoffelen 2016; Buzsaki, G., C. A. Anastassiou, and C. Koch. 2012; Henrie and Shapley 2005; Super and Roelfsema 2005; Kajikawa and Schroeder 2011; Chestek et al. 2020). In our paper, we used phase locking value calculations to demonstrate that MUA power increases with stimuli and MUA events show phase locking to LFP (**Figure 3**) for both organoid and cortex channels. To support our claim of ***local generation of MUA signals***, we have performed additional MUA analysis for the recordings shown in **Figure 3**, demonstrating that MUA events recorded in one channel were unique to that channel (**Figure 3c-d** and **Supplementary Figure 4**, included below) and they cannot originate from neighboring channels. In **Figure 3c** and **Supplementary Figure 4b**, we show event-triggered average MUA traces. Event-triggered MUA averages are computed (technique adopted from Khodagholy et al. 2015) by taking a target channel's (target channel indicated in the title of subfigure in **Supplementary Figure 4b**) MUA event times and, for all 16 channels, computing the average MUA waveform from 1 ms before to 2 ms after each of the target channel's events. If the same neural events are picked up by multiple channels, then similar MUA averages would appear across several, nearby channels. For example, if MUA spikes recorded by organoid channels were generated by the cortex, the same MUA events would appear across nearby cortex channels. And if the channels record locally generated neural activity, then the MUA spike will only appear in the target channel and all other channels will average to zero. The results show that MUA activity recorded from the organoid are completely localized spatially and specific to each channel. Similarly, MUA recorded by cortex channels are local and do not show any spread across other channels. In **Figure 3c** shows four representative channels from the organoid (O1, O4) and cortex (C2, C7) overlaid in the same plot. **Supplementary Figure 4b** shows event-triggered MUA for each channel separately, clearly demonstrating that for the time points when MUA was detected for a certain channel, event-triggered MUA averages to zero for all other channels. Furthermore, In **Figure 3d** and **Supplementary Figure 4c**, we count the number of overlapping events across channels after 1-ms binning. Our results show that there is almost no co-occurrence of events. Finally, we compute the statistical significance of small numbers of event overlaps between the organoid channels (O3, O4, O6) and the cortex channels (C5, C6, C7) by circularly shuffling organoid-cortex channel pairings. **Supplementary Figure 4d** shows that the number of MUA event

overlaps between the organoid and cortex channels fall within the chance range. Overall, the recordings in one channel do not appear to overlap with any surrounding channels, proving that ***MUA records local neural firing and MUA recordings from the organoid are generated locally by the organoid neurons and cannot originate from the surrounding cortex channels.*** Combining this with the data in our paper showing a phase locking of MUA events to ***LFP of the same channel (Figure 3f-g)*** when a visual stimulus was applied, we demonstrate that our LFP recordings with transparent electrodes are primarily recording locally generated neural activity (Buzsaki, Anastassiou, and Koch 2012). To clearly address the concerns raised by the reviewer, we revised the manuscript to include the below statements explicitly stating that ***we verified the locality of MUA and LFPs.*** We also included the new MUA analysis as **Figure 3c-d** and **Supplementary Figure 4**, added additional citations supporting the locality of MUA, and added our new analysis on event overlaps to the Methods section.

In the Results section “Multi-unit activity is modulated in presence of sensory stimuli“:

“To verify that the MUA recordings were localized spatially and unique to each channel, we evaluated event-triggered MUA traces (Khodagholy et al. 2015). Event-triggered MUA averages were computed by taking a target channel’s MUA event times and, for all 16 channels, computing the average MUA waveform from 1 ms before to 2 ms after each of the target channel’s events. If the same neural events were picked up by multiple channels, then similar MUA averages would appear across several, nearby channels. And if the channels recorded independent neural activity, then the MUA event deflection would only appear in the target channel and all other channels would average to zero. Our results showed that MUA events recorded from each channel overlaying the organoid or cortex were local and did not show any spread across other channels (**Figure 3c, Supplementary Figure 4b**). Counting the number of overlapping events between channels after 1-ms binning, we saw that that there was almost no co-occurrence of events (**Figure 3d, Supplementary Figure 4c**). The number of MUA event overlaps between the organoid and cortex channels were not statistically significant and fell within the chance range (**Supplementary Figure 4d**), supporting that MUA recordings originate from local neural firing and therefore recordings from electrodes overlaying the organoid are generated locally by the organoid neurons and cannot originate from the surrounding cortex channels.”

“Koch et al. (Buzsaki, Anastassiou, and Koch 2012) suggested that “verification of the local nature of the signal always requires the demonstration of a correlation between the LFP and local neuronal firing.” Therefore, we performed phase locking analysis to investigate local nature of LFPs. Demonstration of phase locking between LFP and MUA, which we verified to be local to channels, supports that our LFP recordings with transparent electrodes are primarily recording locally generated neural activity.”

In the Discussion section:

“While LFP signals represent spatial integration of extracellular potentials generated by several mechanisms including synaptic transmission, action potentials, intrinsic currents and ephaptic conduction, its locality and spatial specificity depends on the frequency range. We verified the locality of LFP by 1) correlating the MUA with LFP, 2) inspecting the time delay of LFP signals, 3) analyzing high-frequency LFP, and 4) performing independent component analysis (ICA) as one of our preprocessing steps. These analyses suggested that LFP and MUA signals recorded by the channels overlapping with the organoid were generated by the activity of the human neurons.”

In the Methods section “Electrophysiology Data Analysis and Statistics”:

“Event triggered MUA averages were computed by taking a target channel’s MUA event times and, for all 16 channels, computing the average MUA waveform from 1 ms before to 2 ms after each of the target channel’s events. MUA event overlap was determined by binning the MUA events into 1-ms bins then counting the number of overlapping events across channels. Significance was calculated by randomly shifting the event train of one channel 10,000 times and counting the number of overlaps against another channel, with p-value as the number of times the shifted overlap count exceeded the non-shifted case count.”

Figure 3. Multi-unit activity of organoid and cortex. (a) Microphotograph of implantation site (red outline) and surrounding cortex. Electrode pads of the graphene microelectrode array are highlighted; channels 1 and 2 are defined as ‘organoid channels’ whereas channels 3 and 4 are

defined as ‘cortex channels.’ (b) Recording of spontaneous MUA from channels 1-4 on 49 dpi. Dots indicate MUA events crossing the -3.5 sdv threshold. (c) Representative examples of event-averaged MUA traces for four channels (O1, O4, C2, and C7) showing the spatial localization of spontaneous MUA events. (d) Count of overlapping events across channels after binning into 1-ms epochs shows almost no overlapping of events. (e) Trial average (6 trials) of rectified MUA activity in response to the first pulse of a 4-s train of 100-ms light pulses at 2 Hz delivered to the contralateral eye. Red numbers indicate the signal-to-noise ratio of the peak response. (f) Average phase locking value (PLV) vs. frequency for channel #2 (organoid channel, top) and channel #4 (cortex channel, bottom). Arrows point to frequency regions where the PLV was higher during stimulation (black) compared to periods without stimulation (gray). Black dots indicate frequencies where PLVs are significantly different (95% bootstrap confidence interval). Theta PLV at 4-6 Hz is further analyzed in panels e and f. (g) Radial histograms of 5-Hz LFP signal phases during MUA events in channels 2 and 4. (h) Color map of the spatial extent of phase locking; stimulus-induced MUA events of both channels show strongest phase locking to theta oscillations closest to the visual cortex (bottom right). Asterisks indicate $p < 0.05$ between episodes with and without stimulation (only shown on the left plot).

Supplementary Figure 4. MUA analysis in three mice to investigate the overlap of signal across channels. (a) Brightfield image of mouse cortex with organoid region outlined in red. Red channels are those overlapping the organoid. (b) Event-averaged MUA traces show that the MUA events are localized spatially (mean \pm sdv). (c) Table showing the number of overlapping events after

binning events into 1 ms windows for a ~100 s spontaneous recording trial. Diagonal shows the number of events detected per channel. Red color channels are channels overlaying the organoid, blue color channels are those overlaying cortex. **(d)** Histograms of the number of overlaps across an organoid channel and cortex channel (shown in plot titles) after circularly shuffling the MUA event trains 10,000 times. P-values were determined by integrating the shuffled counts from the overlap count of the non-shuffled case (n) in panel c to infinity. Large p-values indicate no significant overlap across channels, supporting that the MUA data is independent across channels.

2. High-Frequency LFP

LFP in the high gamma frequency range (>100 Hz) has been suggested to be strongly correlated with local spiking activity (Scheffer-Teixeira et al. 2013; Einevoll et al. 2013; Petterson and Einevoll 2008; Rasch et al. 2008; Ray et al. 2008; Buszaki, Anastassiou, and Koch 2012). In studies with macaque monkeys, high gamma power of LFPs was found to be highly correlated with spiking activity of neurons (Rasch et al. 2008; Berens et al. 2008). High gamma activity is contributed by local spiking mainly due to attenuation of high frequency signals in brain tissue (Buszaki, Anastassiou, and Koch 2012). Therefore, we included high frequency gamma range activity in our LFP analysis (**Figure 2e**) to ensure that our analysis of cortex vs. organoid signals was accurate. We added below discussions and references to the manuscript to clarify our point:

In the Results section “Organoids generate neural responses to sensory stimuli”:

“LFP in the high gamma frequency range (>100 Hz) has been suggested to be strongly correlated with local spiking activity (Scheffer-Teixeira et al. 2013; Einevoll et al. 2013; Petterson and Einevoll 2008; Rasch et al. 2008; Ray et al. 2008; Buszaki, Anastassiou, and Koch 2012). In studies with macaque monkeys, high gamma power of LFPs was found to be highly correlated with spiking activity of neurons (Rasch et al. 2008; Berens et al. 2008). Our recordings also showed high gamma range activity in both organoid and cortex channels (**Figure 2e**) suggesting that the activity originated from the underlying organoid or cortical tissue and not volume conducted signals.”

3. Independent Component Analysis

Real electrophysiological signal traveling between cortical regions will have a time delay based on underlying axonal projections (Mohajerani et al. 2013) whereas signal propagation due to volume conduction referring to the currents flowing in the tissues surrounding active neuronal sources is considered to be instantaneous across channels (Bastos and Schoffelen 2016; Sejnowski et al. 1996; Jung et al. 1998). ICA is a method that statistically performs blind source separation of the common signals contributing to channel recordings, such as volume conducted signals (Sejnowski et al. 1996; Jung et al. 1998). Therefore, the first step of our pre-processing algorithm is to perform ICA on the raw signal and manually remove any components that have the same influence across all electrode channels. This step ensures that common noise and artifacts due to volume conduction (and other environmental factors such as electrical noise and mouse motion) are removed before signal analysis. We have included the below discussions to emphasize our use of ICA to remove volume conducted signals to our Results and Methods sections respectively.

In the Results section “Organoids generate neural responses to sensory stimuli”:

“Broadband LFP signals (low-pass filtered, 250 Hz) are the spatial summation of extracellular potentials generated by several mechanisms including synaptic transmission, action potentials, intrinsic currents and volume conduction. Relative contribution of these

mechanism and spatial locality and specificity of LFPs depend on the frequency. Volume conduction refers to the currents flowing in the tissues surrounding active neuronal sources and its effects are considered instantaneous across channels (Bastos and Schoffelen 2016; Sejnowski et al. 1996; Jung et al. 1998). Therefore, to eliminate the probable volume conducted signal shared between channels, we performed independent component analysis (ICA) during our data pre-processing. ICA is a method that statistically performs blind source separation of the common signals contributing to channel recordings, such as volume conducted signals (Sejnowski 1996; Jung 1998). By manually removing the signal components shared across all channels, we ensured that noise and artifacts due to volume conduction (and other environmental factors such as electrical noise and mouse motion) were removed before signal analysis.”

In the Methods section “Electrophysiology Data Analysis and Statistics”:

“Data was pre-processed to remove common artifacts from awake mouse motion, 60 Hz power line noise, and shared volume conducted signals using Independent Component Analysis with the *jadeR* algorithm in EEGLab (Delorme and Makeig 2004; Delorme, Sejnowski, and Makeig 2007; Stejnowski 1996; Jung 1998).”

4. LFP Time Delays

Effects of volume conduction in the neural tissues are instantaneous across channels (Bastos and Schoffelen 2016; Sejnowski et al. 1996; Jung et al. 1998) in contrast to time delays observed in propagation of real biological signal through axonal projections. That instantaneous effect of volume conduction is always experimentally observed when electrical stimulation is employed during neural recordings, where volume conducted stimulation artifacts appear simultaneously across all the channels. However, we do not observe any instantaneous signals in our LFP recordings. Our observation of differing time delays in the LFP signal peaks ranging from 35 to 43 ms across channels (**Figure 2d**) supports that our recorded LFP responses were due to real physiological changes in the organoid or cortical cells underlying each channel (Mohajerani et al. 2013). We have added the below discussion and citations to our manuscript to strengthen this point.

In the Results section “Organoids generate neural responses to sensory stimuli”:

“Going further, electrophysiological signal traveling between cortical regions will have a time delay based on underlying axonal projections (Mohajerani 2013). Therefore, to test whether LFP responses were locally generated by the organoid as a result of biological signals propagating via synaptic connections from the cortex or they were signals detected as a result of volume conduction, we examined the relative time-course of LFPs across recording sites. We observed a propagation pattern starting at the area closest to visual cortex (bottom right) and expanding to the implanted organoid area. We quantified this propagation towards the organoid region using the amplitude and onset of the first LFP peak of the trial average (**Figure 2d**). Peak LFP amplitudes close to visual cortex were around 200 μ V and occurred 35 ms after stimulation onset. The channels overlapping with the organoids also detected LFPs with amplitudes of \sim 50 μ V that occurred 43 ms after stimulus onset. The 35-ms latency in vicinity to V1 matches previous observations in intact mouse visual cortex (Land et al. 2013; Kuroki et al. 2018; Lopez et al. 2002), and a 10-20-ms longer delay is consistent with previously observed latencies for the intact RSC region (Funayama et al. 2015).”

2. Visualization of the organoid with 2p imaging

The authors used in vivo real-time two-photon imaging to visualize the implanted organoids. I wondered why the authors did not use fluorescent markers to track transplanted organoids, as in the study by Rios and Clevers (Nature Methods, 2018). It would be much easier to trace the evolution of organoids with a fluorescence maker and will provide direct evidence for morphological fusion between the organoid and mouse cortex.

Our response: We thank the reviewer for this comment. We did not include fluorescent markers in our study for the reasons mentioned in response to comment 1. Unfortunately, repeating all the in vivo organoid transplantation experiments to include fluorescent markers would require at least a year and would not further strengthen our main focus of studying **functional electrophysiological responses evoked by external stimuli from human brain organoids implanted in vivo** and **a novel experimental paradigm for in vivo longitudinal multimodal recordings** of human using **transparent graphene microelectrode arrays**. To address the question of morphological fusion, we include performed staining of human nucleoli, showing organoid cells migrating into mouse cortex (**Figure 5b and Supplementary Figure 7**), 2-photon images of vascularization of organoids (**Figure 5a**), and CD31, showing vascularization of the organoid by mouse blood vessels (**Figure 5c**).

Additionally, to improve our claims of morphological fusion of organoids to mouse cortex, we performed post-mortem immunofluorescence staining of synaptophysin (Syn) and post-synaptic density markers along with human nucleoli (NM-95) and human cytoplasm (STEM121) markers to show the synaptic development within organoids and an overlap of synapses between organoids and mouse cortex. We used our fixed brain slices (extracted 11 weeks post-organoid implantation, directly after the final electrophysiology recording) for the immunofluorescence staining. We included the new results as **Figure 6a-c**, where we examined the co-localization of Syn (red) and PSD95 (green) with human cells (white) along the boundary (delineated in pink) of the transplanted organoid (Org) and visual cortex (Visual), the boundary (delineated in pink) of the organoid and retrosplenial cortex (RSC), and within the corpus callosum (CC). We observed clear co-localization of organoid cells with the pre- and post-synaptic markers at the boundary of the mouse visual cortex (**Figure 6a**, arrowheads), retrosplenial cortex (**Figure 6b**, arrowheads), and corpus callosum (**Figure 6c**, arrowheads), suggesting synaptic connectivity.

Moreover, we investigated whether the synaptophysin within the organoid was of mouse or human origin and evaluated bi-directional synaptic connections between xenografted organoids and host mouse cortex with post-mortem immunofluorescence analysis. To the best of our knowledge, there is no mouse-specific synaptophysin marker, so we performed triple staining for human cytoplasm (STEM121), Syn, and human-specific synaptophysin (hSyn) (n=2). Focusing on the overlap of Syn and hSyn, we could determine which presynaptic puncta were of human origin. The remaining puncta that labeled positive for Syn but negative for hSyn were counted as presynaptic puncta of mouse origin. **Figure 6d** and **e** show presynaptic puncta and organoid neuronal projections inside the organoid 100 μm from visual cortex boundary and at the organoid center, respectively. We observed mouse presynaptic puncta (hollow arrowheads) colocalized nearby the organoid projections both at the boundary (**Figure 6d**) and the center (**Figure 6e**) of the organoid, suggesting mouse neurons formed pre-synaptic connections to the organoid. Quantifying the density of human and mouse presynaptic puncta, we observed a significantly greater density of mouse puncta at the boundary compared to center of the organoid, likely due to the proximity to mouse cortex, and a greater density of human puncta at the center of the organoid compared to the boundary, likely due to the dispersion of human cells as they morphologically integrated with mouse cortex (**Figure 6f**). We also observed human presynaptic puncta and STEM121 staining in mouse visual cortex near the outer edge of the organoid (**Supplementary Figure 8**). Our observations of mouse presynaptic puncta in the organoid and

human presynaptic puncta in the mouse cortex support that by the time of the final recording session, organoid and mouse visual cortex had made bi-directional synaptic connections. Mouse presynaptic puncta in the organoid provides evidence for the synaptic connectivity needed by the organoid to generate functional responses to visual stimuli, as observed in electrophysiological recordings. These new results are included as **Figure 6** and **Supplementary Figure 8** and the following discussions were added to the manuscript.

In the Results section “Multi-unit activity is modulated in presence of sensory stimuli”:

“These synaptic connections were later verified using post-mortem immunofluorescence staining for human cytoplasm (STEM121), human nucleoli (NM-95), post-synaptic density (PSD95), and human- (hSyn) and non-species-specific synaptophysin (Syn) markers.”

In the Results section “Organoid grafts are vascularized by the host and integrate with surrounding cortex”:

“Finally, in order to investigate synaptic connectivity between organoid and cortex, we performed immunofluorescence staining for human nucleoli (NM-95), pre-synaptic vesicle protein synaptophysin (Syn), and post-synaptic densities (PSD95). In **Figure 6a-c**, we examined the co-localization of Syn (red) and PSD95 (green) with human cells (white) along the boundary (delineated in pink) of the transplanted organoid (Org) and visual cortex (Visual), the boundary (delineated in pink) of the organoid and retrosplenial cortex (RSC), and within the corpus callosum (CC). We observed clear co-localization of organoid cells with the pre- and post-synaptic markers at the boundary of the mouse visual cortex (**Figure 6a**, arrowheads), retrosplenial cortex (**Figure 6b**, arrowheads), and corpus callosum (**Figure 6c**, arrowheads), suggesting synaptic connectivity. Moreover, we investigated whether the synaptophysin within the organoid was of mouse or human origin and evaluated bi-directional synaptic connections between xenografted organoids and host mouse cortex with post-mortem immunofluorescence analysis. To the best of our knowledge, there is no mouse-specific synaptophysin marker, so we performed triple staining for human cytoplasm (STEM121), Syn, and human-specific synaptophysin (hSyn) (n=2). Focusing on the overlap of Syn and hSyn, we could determine which presynaptic puncta were of human origin. The remaining puncta that labeled positive for Syn but negative for hSyn were counted as presynaptic puncta of mouse origin. **Figure 6d** and **e** show presynaptic puncta and organoid neuronal projections inside the organoid 100 μm from visual cortex boundary and at the organoid center, respectively. We observed mouse presynaptic puncta (hollow arrowheads) colocalized nearby the organoid projections both at the boundary (**Figure 6d**) and the center (**Figure 6e**) of the organoid, suggesting mouse neurons formed pre-synaptic connections to the organoid. Quantifying the density of human and mouse presynaptic puncta, we observed a significantly greater density of mouse puncta at the boundary compared to center of the organoid, likely due to the proximity to mouse cortex, and a greater density of human puncta at the center of the organoid compared to the boundary, likely due to the dispersion of human cells as they morphologically integrated with mouse cortex (**Figure 6f**). We also observed human presynaptic puncta and STEM121 staining in mouse visual cortex near the outer edge of the organoid (**Supplementary Figure 8**). Our observations of mouse presynaptic puncta in the organoid and human presynaptic puncta in the mouse cortex support that by the time of the final recording session, organoid and mouse visual cortex had made bi-directional synaptic connections. Mouse presynaptic puncta in the organoid provides

evidence for the synaptic connectivity needed by the organoid to generate functional responses to visual stimuli, as observed in electrophysiological recordings.”

Figure 5. In vivo imaging of organoid vascularization and post-mortem immunohistochemical analysis. (a) Depth projections (0-650 μm) of image stacks acquired using two-photon microscopy (1240 nm excitation wavelength, 512x512 pixel, 3 μm step size) after injection of the intravascular

tracer Alexa 680-Dextran. The organoid implant region is outlined in red in the overview image (left) and vasculature of organoid transplant site (center) and mouse cortex (right) are shown. The yellow boxes with thicker outlines highlight representative organoid regions shown in greater detail in the middle panels and the yellow boxes with thin outlines highlight mouse cortical regions shown in greater detail in the right panels. **(b)** Immunostaining of a 5- μ m coronal section with NM-95 antibodies detecting human nucleoli. The implant region (center) contains mainly human cells. Organoid-derived (human) cells (arrows) are also present in surrounding mouse cortex (right). **(c)** Immunostaining of a 5- μ m coronal section with CD31 antibodies detecting endothelial cells. Staining indicates vascularization of the implanted organoid (center, arrows) although at lower density compared to surrounding cortex (right, arrows). **(d)** Immunostaining with NeuN antibodies detecting neuronal nuclei indicates cells in the implant regions which are neurons. Note that NeuN antibodies show weaker staining of human neuronal nuclei (center, arrows) compared to mouse neuronal nuclei (right, arrows). Sections shown in panels b-d were counterstained with hematoxylin; primary antibodies were detected with horseradish-peroxidase-coupled secondary antibodies and diaminobenzidine as chromogenic substrate. **(e)** Composition analysis of organoid using NeuN, NM-95, Ki67, and Olig2 within organoid and comparison to contralateral cortical tissue.

Figure 6. Confocal microscopy of immunofluorescence staining for antibodies against human nucleoli (NM-95), human cytoplasm (STEM121), Synaptophysin (Syn; pre-synaptic terminal vesicle protein), and post-synaptic density (PSD-95) in a region with organoid implant taken from a mouse directly after the last electrophysiology recording at 11 weeks post-implantation. DAPI (blue) was used as counterstain for cell nuclei. **(a)** Co-localization of Syn (red) and PSD-95 (green) with human cells along the boundary (delineated in pink) of the organoid implant (Org) and visual cortex (Visual, arrowheads). **(b)** Human cells were observed surrounded by Syn, PSD-95, and mouse cells (arrowheads) in a region at the organoid implant and retrosplenial cortex (RSC) boundary (delineated in pink). **(c)** Human cells (arrowheads) were observed surrounded by PSD-95 and Syn, traveling along the corpus callosum (CC). The boundary between CC and visual cortex is delineated in pink. Note the smaller density of Syn and PSD-95, which is characteristic of the corpus callosum due to longer myelinated axonal projections and less synaptic terminals.

(d) Puncta positive for Syn (green) and negative for hSyn (red) were defined as mouse pre-synaptic terminals. The number of mouse pre-synaptic puncta was high at the boundary of the organoid implant (d, hollow arrowheads) but still existed at the center of the implant (e, hollow arrowheads). Solid arrowheads in (d-e) indicate puncta positive for both hSyn and Syn, representing human pre-synaptic terminals. (f) The density of human pre-synaptic terminals significantly increased towards the center of the organoid (top bar chart) and the density of mouse pre-synaptic terminals (i.e. +Syn/-hSyn puncta) significantly decreased towards the center of the organoid (bottom bar chart) (*p < 0.01 in two-sample t-test).

Supplementary Figure 7. Human (NM-95-positive) cells were observed at the implantation site (green arrow) and further away from the implantation site (a, b); we detected individual human cells along corpus callosum up to ~4 mm away from the implantation site (b).

Supplementary Figure 8. Puncta that labeled positive for both Syn (green) and hSyn (red) were counted as presynaptic puncta of human origin (yellow, arrowheads). We observed a presence of STEM121 (white) and human presynaptic puncta (yellow, arrowheads) within regions of visual cortex, supporting the organoid extended axonal connections towards and into mouse visual cortex.

Minor

1. Did you try different anesthesia, other than isoflourane?
2. Have you tried any anesthesia other than isoflourane? MoA is different for each anesthetic, so if different anesthetics produce different consequences, it may provide useful insight into organoid transplantation research.

Our response: We did not try different anesthesia other than isoflurane but agree that since different anesthetics have differing mechanisms of action and effects on brain activity (Michelson and Kozai 2018; Hayton, Kriss, and Muller 1999), trying different anesthetics would be an interesting topic of study in a future paper. We added the below comment to the Discussion section of the manuscript.

“Anesthetics differ greatly in their mechanisms of action and influence on cortical activity (Michelson and Kozai 2018; Hayton, Kriss, and Muller 1999) and investigating the influence of other anesthetics on xenografted organoids would be an interesting future application of this methodology.”

3. In Supplementary Figure 3, the change in MUA spike rate over time is markedly different in 4 mice. In particular, the data from Mouse 3 stands out. What's the reason?

Our response: We checked the histology results from mouse 3 and found that the organoid at 11 weeks post-implantation was much smaller compared to other mice, which might explain why the MUA event rate is different compared to the other mice. The organoid cells in that mouse may have been less healthy post-implantation and thus retained a low, decreasing firing rate over the course of the experiment.

Reviewer #2 (Remarks to the Author):

The paper by Wilson et al., describes new method combining transplantation of cortical organoids derived from human iPSC and transparent graphene microelectrodes. The study shows that grafted organoids reveal that peripheral stimulation evokes electrophysiological responses in the organoids. Although, this studies technically very sound and advances our capacity to monitor activity of transplanted organoids it has serious shortcomings. Firstly, it is very poor in morphological representation of the transplanted organoids and do not show even neurons, only single stained human cells with unknown phenotype. Authors need to demonstrate what is the neuronal survival and composition of the organoids at a time of recording and also cytoarchitecture of the grafted and surrounding tissue.

Our response: We thank Reviewer 2 for their recognition of the soundness of our study and potential for using our technology for organoid investigation. Following the reviewer's suggestion on morphological representation, we show neuronal survival in **figure 5d** using NeuN staining for neuronal nucleoli. To further address this comment, we performed additional post-mortem immunohistochemical composition analysis of NeuN, NM-95, Ki67, and Olig2 to determine the composition of organoids upon the time of last recording (8-11 weeks post-implantation). We found that the composition of organoids was ~48% neurons, ~7% oligodendrocytes, ~5% Ki67+ cells, and ~82% human nucleoli+ cells (n=3). This new information has been added to the manuscript as **Figure 5e** and we've added the below discussion to elaborate our findings:

In the Results section "Organoid grafts are vascularized by the host and integrate with surrounding cortex":

"NeuN co-staining with hematoxylin revealed that ~48% of the cells in the organoid graft had a neuronal phenotype at the final time of recording (**Figure 5d-e**)... Additional staining for NM-95, proliferating cells (Ki67), and oligodendrocytes (Olig2) yielded the organoid composition as consisting of ~82% human cells, ~5% proliferating cells, and ~7% oligodendrocytes (n=3, **Figure 5e**)."

Furthermore, we performed additional staining of pre- and post-synaptic markers PSD-95 and synaptophysin and found a presence of PSD-95 in the organoid regions (**Figure 6a-c**, co-localizing with NM-95 staining). This data supports that many of the organoid cells had a neuronal fate because PSD-95 is only found in excitatory neuron synapses (El-Husseini et al. 2000). We have added **Figure 6** to our manuscript with figures of the pre- and post-synaptic immunofluorescence staining and added below discussions regarding the appearance of PSD-95 in the organoid to support the neuronal fate of organoid cells.

In the Results section "Organoid grafts are vascularized by the host and integrate with surrounding cortex":

"The neuronal phenotype of organoid cells was further supported by the detection of PSD-95 in the organoid region which is a post-synaptic density marker found solely in excitatory neuron synapses (**Figure 6**) (El-Husseini et al. 2000)."

Figure 5. In vivo imaging of organoid vascularization and post-mortem immunohistochemical analysis. (a) Depth projections (0-650 μm) of image stacks acquired using two-photon microscopy (1240 nm excitation wavelength, 512x512 pixel, 3 μm step size) after injection of the intravascular tracer Alexa 680-Dextran. The organoid implant region is outlined in red in the overview image (left) and vasculature of organoid transplant site (center) and mouse cortex (right) are shown. The yellow boxes with thicker outlines highlight representative organoid regions shown in greater detail in the middle panels and the yellow boxes with thin outlines highlight mouse cortical regions shown in greater detail in the right panels. (b) Immunostaining of a 5- μm coronal section with NM-

95 antibodies detecting human nucleoli. The implant region (center) contains mainly human cells. Organoid-derived (human) cells (arrows) are also present in surrounding mouse cortex (right). **(c)** Immunostaining of a 5- μ m coronal section with CD31 antibodies detecting endothelial cells. Staining indicates vascularization of the implanted organoid (center, arrows) although at lower density compared to surrounding cortex (right, arrows). **(d)** Immunostaining with NeuN antibodies detecting neuronal nuclei indicates cells in the implant regions which are neurons. Note that NeuN antibodies show weaker staining of human neuronal nuclei (center, arrows) compared to mouse neuronal nuclei (right, arrows). Sections shown in panels b-d were counterstained with hematoxylin; primary antibodies were detected with horseradish-peroxidase-coupled secondary antibodies and diaminobenzidine as chromogenic substrate. **(e)** Composition analysis of organoid using NeuN, NM-95, Ki67, and Olig2 within organoid and comparison to contralateral cortical tissue.

Secondly, authors fail to demonstrate which brain structures (sub-cortical and cortical) make synaptic connections with transplanted organoid neurons and whether those grafted neurons establish synaptic connections of the host brain. These issues have been successfully addressed in recent studies using rabies virus-based retrograde labeling in combination with other methods to demonstrate functional afferent and efferent connection of the cortical neurons derived from human iPSCs and transplanted in lesioned cortex. Unfortunately, authors do not even mention these papers. I believe when authors discuss functional connectivity and integration of grafted organoids it is of great importance to demonstrate their connectivity to the host brain in order to fully understand the usefulness of present approach for modeling and exploring human brain development and dysfunction.

Our response: We thank the reviewer for suggesting rabies tracing methodology. To address synaptic connection, we did not perform rabies tracing in the organoids as the method for iPSC organoids *in vivo* was not yet established. We looked through the literature for the papers Reviewer 2 mentioned on rabies virus tracing. However, the papers we found trace iPSC extensions *in vivo* (Lu et al. 2014; Cunningham et al. 2014) or organoid axonal extensions *in vitro* (Miura et al. 2020; Anderson et al. 2020) and thus don't exactly match our study. To our understanding, there isn't yet an established protocol for labeling transplanted iPSC-derived organoids in mouse cortices and thus might take another year to establish ourselves. However, we added below sentence to the Discussion section to recommend this method as future work and cited these papers. We would appreciate if the reviewer could point out any references we might be missing.

In the Discussion section:

“Another future direction for this technology could be to take advantage of the electrode transparency by incorporating calcium imaging to visualize spiking activity in organoid neurons or rabies viral retrograde tracing of axonal projections between organoid and mouse cortex as others have demonstrated with iPSCs *in vivo* (Lu et al. 2014; Cunningham et al. 2014) and spheroids *in vitro* (Miura et al. 2020; Anderson et al. 2020). These methods were not established for our specific cell lines at the time of this experiment but could be developed for future use.”

As an alternative method to support our manuscript's claim for synaptic connectivity, we performed additional synaptic histology, looking at the co-localization of antibodies post-synaptic density (PSD95), synaptophysin (Syn), human cytoplasm (STEM121), human nucleoli (NM-95),

and DAPI in post-mortem tissue which have been used in previous papers analyzing iPSC synaptic connectivity (Mansour et al. 2018; Lu et al. 2014; Cunningham et al. 2014; Miura et al. 2020; Giandomenico et al. 2019; Shi et al. 2020). We used our fixed brain slices (extracted 11 weeks post-organoid implantation, directly after the final electrophysiology recording) and performed immunofluorescence. We included the new results as **Figure 6a-c**, where we examined the co-localization of Syn (red) and PSD95 (green) with human cells (white) along the boundary (delineated in pink) of the transplanted organoid (Org) and visual cortex (Visual), the boundary (delineated in pink) of the organoid and retrosplenial cortex (RSC), and within the corpus callosum (CC). We observed clear co-localization of organoid cells with the pre- and post-synaptic markers at the boundary of the mouse visual cortex (**Figure 6a**, arrowheads), retrosplenial cortex (**Figure 6b**, arrowheads), and corpus callosum (**Figure 6c**, arrowheads), suggesting synaptic connectivity.

Moreover, we performed immunofluorescence with human-specific synaptophysin (hSyn) and human cytoplasmic protein marker (STEM121) to investigate whether the synaptophysin within the organoid was of mouse or human origin. The hSyn we used was previously demonstrated to support functional connectivity between mouse cortex and human organoids by our coauthors Dr. Mansour and Dr. Gage (Mansour et al. 2018). STEM 121 has been used in several publications to show axonal projections and morphological integration of human organoids in mice and rat brains (Palma-Tortosa et al. 2020; Pham et al. 2018; Dong et al. 2021). Despite exhaustive searches in the literature and of vendor websites, we could not find a mouse-specific synaptophysin marker. Thus, to support our claim that mouse cortex made pre-synaptic connections to the organoid, within the organoid we performed a double stain with non-specific synaptophysin (Syn) and hSyn. Focusing on the overlap of Syn and hSyn, we could determine which presynaptic puncta were of human origin. The remaining puncta that labeled positive for Syn but negative for hSyn were counted as presynaptic puncta of mouse origin. **Figure 6d** and **e** show presynaptic puncta and organoid neuronal projections inside the organoid 100 μm from visual cortex boundary and at the organoid center, respectively. We observed mouse presynaptic puncta (hollow arrowheads) colocalized nearby the organoid projections both at the boundary (**Figure 6d**) and the center (**Figure 6e**) of the organoid, suggesting mouse neurons formed pre-synaptic connections to the organoid. Quantifying the density of human and mouse presynaptic puncta, we observed a significantly greater density of mouse puncta at the boundary compared to center of the organoid, likely due to the proximity to mouse cortex, and a greater density of human puncta at the center of the organoid compared to the boundary, likely due to the dispersion of human cells as they morphologically integrated with mouse cortex (**Figure 6f**). We also observed human presynaptic puncta and STEM121 staining in mouse visual cortex near the outer edge of the organoid (**Supplementary Figure 8**). Our observations of mouse presynaptic puncta in the organoid and human presynaptic puncta in the mouse cortex support that by the time of the final recording session, organoid and mouse visual cortex had made bi-directional synaptic connections. Mouse presynaptic puncta in the organoid provides evidence for the synaptic connectivity needed by the organoid to generate functional responses to visual stimuli, as observed in electrophysiological recordings. These results are included as **Figure 6**, **Supplementary Figure 8**, and the following discussions were added to the manuscript.

In the Results section “Multi-unit activity is modulated in presence of sensory stimuli”:

“These synaptic connections were later verified using post-mortem immunofluorescence staining for human cytoplasm (STEM121), human nucleoli (NM-95), post-synaptic density (PSD95), and human- (hSyn) and non-species-specific synaptophysin (Syn) markers.”

In the Results section “Organoid grafts are vascularized by the host and integrate with surrounding cortex”:

“Finally, in order to investigate synaptic connectivity between organoid and cortex, we performed immunofluorescence staining for human nucleoli (NM-95), pre-synaptic vesicle protein synaptophysin (Syn), and post-synaptic densities (PSD95). In **Figure 6a-c**, we examined the co-localization of Syn (red) and PSD95 (green) with human cells (white) along the boundary (delineated in pink) of the transplanted organoid (Org) and visual cortex (Visual), the boundary (delineated in pink) of the organoid and retrosplenial cortex (RSC), and within the corpus callosum (CC). We observed clear co-localization of organoid cells with the pre- and post-synaptic markers at the boundary of the mouse visual cortex (**Figure 6a**, arrowheads), retrosplenial cortex (**Figure 6b**, arrowheads), and corpus callosum (**Figure 6c**, arrowheads), suggesting synaptic connectivity. Moreover, we investigated whether the synaptophysin within the organoid was of mouse or human origin and evaluated bi-directional synaptic connections between xenografted organoids and host mouse cortex with post-mortem immunofluorescence analysis. To the best of our knowledge, there is no mouse-specific synaptophysin marker, so we performed triple staining for human cytoplasm (STEM121), Syn, and human-specific synaptophysin (hSyn) (n=2). Focusing on the overlap of Syn and hSyn, we could determine which presynaptic puncta were of human origin. The remaining puncta that labeled positive for Syn but negative for hSyn were counted as presynaptic puncta of mouse origin. **Figure 6d** and **e** show presynaptic puncta and organoid neuronal projections inside the organoid 100 μm from visual cortex boundary and at the organoid center, respectively. We observed mouse presynaptic puncta (hollow arrowheads) colocalized nearby the organoid projections both at the boundary (**Figure 6d**) and the center (**Figure 6e**) of the organoid, suggesting mouse neurons formed pre-synaptic connections to the organoid. Quantifying the density of human and mouse presynaptic puncta, we observed a significantly greater density of mouse puncta at the boundary compared to center of the organoid, likely due to the proximity to mouse cortex, and a greater density of human puncta at the center of the organoid compared to the boundary, likely due to the dispersion of human cells as they morphologically integrated with mouse cortex (**Figure 6f**). We also observed human presynaptic puncta and STEM121 staining in mouse visual cortex near the outer edge of the organoid (**Supplementary Figure 8**). Our observations of mouse presynaptic puncta in the organoid and human presynaptic puncta in the mouse cortex support that by the time of the final recording session, organoid and mouse visual cortex had made bi-directional synaptic connections. Mouse presynaptic puncta in the organoid provides evidence for the synaptic connectivity needed by the organoid to generate functional responses to visual stimuli, as observed in electrophysiological recordings.”

Figure 6. Confocal microscopy of immunofluorescence staining for antibodies against human nucleoli (NM-95), human cytoplasm (STEM121), Synaptophysin (Syn; pre-synaptic terminal vesicle protein), and post-synaptic density (PSD-95) in a region with organoid implant taken from a mouse directly after the last electrophysiology recording at 11 weeks post-implantation. DAPI (blue) was used as counterstain for cell nuclei. **(a)** Co-localization of Syn (red) and PSD-95 (green) with human cells along the boundary (delineated in pink) of the organoid implant (Org) and visual cortex (Visual, arrowheads). **(b)** Human cells were observed surrounded by Syn, PSD-95, and mouse cells (arrowheads) in a region at the organoid implant and retrosplenial cortex (RSC) boundary (delineated in pink). **(c)** Human cells (arrowheads) were observed surrounded by PSD-95 and Syn, traveling along the corpus callosum (CC). The boundary between CC and visual cortex is delineated in pink. Note the smaller density of Syn and PSD-95, which is characteristic of the corpus callosum due to longer myelinated axonal projections and less synaptic terminals. **(d)** Puncta positive for Syn (green) and negative for hSyn (red) were defined as mouse pre-

synaptic terminals. The number of mouse pre-synaptic puncta was high at the boundary of the organoid implant (**d**, hollow arrowheads) but still existed at the center of the implant (**e**, hollow arrowheads). Solid arrowheads in (d-e) indicate puncta positive for both hSyn and Syn, representing human pre-synaptic terminals. (**f**) The density of human pre-synaptic terminals significantly increased towards the center of the organoid (top bar chart) and the density of mouse pre-synaptic terminals (i.e. +Syn/-hSyn puncta) significantly decreased towards the center of the organoid (bottom bar chart) (* $p < 0.01$ in two-sample t-test).

Supplementary Figure 8. Puncta that labeled positive for both Syn (green) and hSyn (red) were counted as presynaptic puncta of human origin (yellow, arrowheads). We observed a presence of STEM121 (white) and human presynaptic puncta (yellow, arrowheads) within regions of visual cortex, supporting the organoid extended axonal connections towards and into mouse visual cortex.

References

- Andersen, J., Revah, O., Miura, Y., Thom, N., Amin, N. D., Kelley, K. W., ... & Paşca, S. P. (2020). Generation of functional human 3D cortico-motor assembloids. *Cell*, *183*(7), 1913-1929.
- Bastos, A. M., & Schoffelen, J. M. (2016). A tutorial review of functional connectivity analysis methods and their interpretational pitfalls. *Frontiers in systems neuroscience*, *9*, 175.
- Berens, P., Keliris, G. A., Ecker, A. S., Logothetis, N. K., & Tolias, A. S. (2008). Feature selectivity of the gamma-band of the local field potential in primate primary visual cortex. *Frontiers in neuroscience*, *2*, 37.
- Buzsáki, G., C. A. Anastassiou, and C. Koch. 2012. 'The origin of extracellular fields and currents-EEG, ECoG, LFP and spikes', *Nat Rev Neurosci*, *13*: 407-20.
- Cunningham, M., Cho, J. H., Leung, A., Savvidis, G., Ahn, S., Moon, M., ... & Chung, S. (2014). hPSC-derived maturing GABAergic interneurons ameliorate seizures and abnormal behavior in epileptic mice. *Cell stem cell*, *15*(5), 559-573.
- Dong, X., Xu, S. B., Chen, X., Tao, M., Tang, X. Y., Fang, K. H., ... & Liu, Y. (2021). Human cerebral organoids establish subcortical projections in the mouse brain after transplantation. *Molecular psychiatry*, *26*(7), 2964-2976.
- Einevoll, G. T., Kayser, C., Logothetis, N. K., & Panzeri, S. (2013). Modelling and analysis of local field potentials for studying the function of cortical circuits. *Nature Reviews Neuroscience*, *14*(11), 770-785.
- El-Husseini, A. E. D., Schnell, E., Chetkovich, D. M., Nicoll, R. A., & Brecht, D. S. (2000). PSD-95 involvement in maturation of excitatory synapses. *Science*, *290*(5495), 1364-1368.
- Giandomenico, S. L., Mierau, S. B., Gibbons, G. M., Wenger, L., Masullo, L., Sit, T., ... & Lancaster, M. A. (2019). Cerebral organoids at the air-liquid interface generate diverse nerve tracts with functional output. *Nature neuroscience*, *22*(4), 669-679.
- Hayton, S. M., Kriss, A., & Muller, D. P. R. (1999). Comparison of the effects of four anaesthetic agents on somatosensory evoked potentials in the rat. *Laboratory animals*, *33*(3), 243-251.
- Henrie, J. A., and R. Shapley. 2005. 'LFP power spectra in V1 cortex: the graded effect of stimulus contrast', *J Neurophysiol*, *94*: 479-90.
- Jung, T. P., Humphries, C., Lee, T. W., Makeig, S., McKeown, M. J., Iragui, V., & Sejnowski, T. J. (1998, September). Removing electroencephalographic artifacts: comparison between ICA and PCA. In *Neural Networks for Signal Processing VIII. Proceedings of the 1998 IEEE Signal Processing Society Workshop (Cat. No. 98TH8378)* (pp. 63-72). IEEE.
- Kajikawa, Y., & Schroeder, C. E. (2011). How local is the local field potential?. *Neuron*, *72*(5), 847-858.
- Khodagholy, D., Gelinas, J. N., Thesen, T., Doyle, W., Devinsky, O., Malliaras, G. G., & Buzsáki, G. (2015). NeuroGrid: recording action potentials from the surface of the brain. *Nature neuroscience*, *18*(2), 310-315.
- Land, R., Engler, G., Kral, A., & Engel, A. K. (2013). Response properties of local field potentials and multiunit activity in the mouse visual cortex. *Neuroscience*, *254*, 141-151.
- Lu, P., Woodruff, G., Wang, Y., Graham, L., Hunt, M., Wu, D., ... & Tuszynski, M. H. (2014). Long-distance axonal growth from human induced pluripotent stem cells after spinal cord injury. *Neuron*, *83*(4), 789-796.
- Lu, Y., Liu, X., Hattori, R., Ren, C., Zhang, X., Komiyama, T., & Kuzum, D. (2018). Ultralow impedance graphene microelectrodes with high optical transparency for simultaneous deep two-photon imaging in transgenic mice. *Advanced functional materials*, *28*(31), 1800002.
- Mansour, A. A., Gonçalves, J. T., Bloyd, C. W., Li, H., Fernandes, S., Quang, D., ... & Gage, F. H. (2018). An in vivo model of functional and vascularized human brain organoids. *Nature biotechnology*, *36*(5), 432-441.

- Michelson, N. J., & Kozai, T. D. (2018). Isoflurane and ketamine differentially influence spontaneous and evoked laminar electrophysiology in mouse V1. *Journal of neurophysiology*, 120(5), 2232-2245.
- Miura, Y., Li, M. Y., Birey, F., Ikeda, K., Revah, O., Thete, M. V., ... & Paşca, S. P. (2020). Generation of human striatal organoids and cortico-striatal assembloids from human pluripotent stem cells. *Nature Biotechnology*, 38(12), 1421-1430.
- Mohajerani, M. H., Chan, A. W., Mohsenvand, M., LeDue, J., Liu, R., McVea, D. A., ... & Murphy, T. H. (2013). Spontaneous cortical activity alternates between motifs defined by regional axonal projections. *Nature neuroscience*, 16(10), 1426-1435.
- Nason, S. R., Vaskov, A. K., Willsey, M. S., Welle, E. J., An, H., Vu, P. P., ... & Chestek, C. A. (2020). A low-power band of neuronal spiking activity dominated by local single units improves the performance of brain-machine interfaces. *Nature biomedical engineering*, 4(10), 973-983.
- Palma-Tortosa, S., Tornero, D., Hansen, M. G., Monni, E., Hajy, M., Kartsivadze, S., ... & Kokaia, Z. (2020). Activity in grafted human iPS cell-derived cortical neurons integrated in stroke-injured rat brain regulates motor behavior. *Proceedings of the National Academy of Sciences*, 117(16), 9094-9100.
- Pettersen, K. H., & Einevoll, G. T. (2008). Amplitude variability and extracellular low-pass filtering of neuronal spikes. *Biophysical journal*, 94(3), 784-802.
- Pham, M. T., Pollock, K. M., Rose, M. D., Cary, W. A., Stewart, H. R., Zhou, P., ... & Waldau, B. (2018). Generation of human vascularized brain organoids. *Neuroreport*, 29(7), 588.
- Rasch, M. J., Gretton, A., Murayama, Y., Maass, W. & Logothetis, N. K. Inferring spike trains from local field potentials. *J. Neurophysiol.* **99**, 1461–1476 (2008).
- Ray, S., Crone, N.E., Niebur, E., Franaszczuk, P.J. and Hsiao, S.S., 2008. Neural correlates of high-gamma oscillations (60–200 Hz) in macaque local field potentials and their potential implications in electrocorticography. *Journal of Neuroscience*, 28(45), pp.11526-11536.
- Scheffer-Teixeira, R., Belchior, H., Leao, R. N., Ribeiro, S., & Tort, A. B. (2013). On high-frequency field oscillations (> 100 Hz) and the spectral leakage of spiking activity. *Journal of Neuroscience*, 33(4), 1535-1539.
- Sejnowski, T. J. (1996). Independent component analysis of electroencephalographic data. In *Advances in Neural Information Processing Systems 8: Proceedings of the 1995 Conference* (Vol. 8, p. 145). MIT press.
- Shi, Y., Sun, L., Wang, M., Liu, J., Zhong, S., Li, R., ... & Wang, X. (2020). Vascularized human cortical organoids (vOrganoids) model cortical development in vivo. *PLoS biology*, 18(5), e3000705.
- Super, H., & Roelfsema, P. R. 2005. Chronic multiunit recordings in behaving animals: advantages and limitations. *Progress in brain research*, 147, 263-282.
- Thunemann, M., Lu, Y., Liu, X., Kılıç, K., Desjardins, M., Vandenberghe, M., ... & Kuzum, D. (2018). Deep 2-photon imaging and artifact-free optogenetics through transparent graphene microelectrode arrays. *Nature communications*, 9(1), 1-12.
- Trujillo, C. A., Gao, R., Negraes, P. D., Gu, J., Buchanan, J., Preissl, S., ... & Muotri, A. R. (2019). Complex oscillatory waves emerging from cortical organoids model early human brain network development. *Cell stem cell*, 25(4), 558-569.

REVIEWER COMMENTS

Reviewer #3 (Remarks to the Author):

The use of transparent graphene electrodes to get a read-out from a transplanted organoid is of interest. Please consider the comments below:

They said, "Spontaneous MUA events remained relatively stable over the eight- to eleven-week experiments, with spiking rates around 2 Hz." In the four animals shown in the supplement they show MUA spike rate in Hz. Given the known variation in firing rates per electrode and the fact that often an electrode can be silent throughout the duration of an experiment, they should show this in a more granular way across the array (not just averaged). They need to be clearer about the spontaneous activity in the organoid before looking at the stimulus-induced activity.

Fig 3b—this figure is hard to see and appears not to fit the filter definition of crossing their threshold. Many putative MUAs are marked where there appears to be a peak that did not cross the threshold as defined by the red line in the figure. Their pre-defined threshold of -3 to -4 times the SD is somewhat generous—six times is more rigorous and one needs to account for the median noise.

The focus on high gamma LFPs necessarily encounters the difficulty of disentangling high gamma from spikes. The minimal unit on their recordings is 1 sec in Fig 2. Not until we get to figure 3e, do we see the higher time resolution now presented in a different context, i.e. phase locking. We need to see the activity at msec resolution for spiking and we need clarification and more detailed labeling of Fig 2e to see the entire power spectrum and specifically the power bands of interest in relation to the entire spectrum. The Morlet wavelets used by the authors is not a substitute for a Fourier transform or some similar computation.

Given the imprecision of the LFP onset time, computing a 10-20 msec delay in reaching the organoid does not imply coherence of the wave (as they seem to be trying to do). Nor is there a statistic to go along with this very wide range of 10-20 msec to show significance vis à vis the null hypothesis. Thus, this statement is not fully supported: "The consistency in delay times with the literature indicates that the observed responses from the electrode located over the organoid are generated by the activity of human organoid..." {italics are mine} More importantly, concluding that the LFPs came from the organoid is problematic even after they exclude volume conduction. And by the way, there is extensive and extraneous background on volume conduction in the results section. In these small samples, volume conduction is unlikely to have major effects and all of that material fits better as supplemental.

The basis of the phase locking in the radial histograms does not include statistical support that this reviewer can find and they do not look very convincing. As they noted earlier there are spontaneous LFPs from which the light responsive ones need to be distinguished.

One would like to see controls that change the stimulus duration and intensity and deliver stimulus to ipsilateral eye.

REVIEWER COMMENTS

We would like to thank Reviewer 3 for kindly reviewing our manuscript and highly appreciate their insightful comments. We have revised the manuscript according to the suggestions and believe that the revisions have improved the paper. Please find below our responses (in regular fonts) to each specific comment (in italic fonts) provided by the reviewer.

Comment 1: *The use of transparent graphene electrodes to get a read-out from a transplanted organoid is of interest. Please consider the comments below:*

They said, "Spontaneous MUA events remained relatively stable over the eight- to eleven-week experiments, with spiking rates around 2 Hz." In the four animals shown in the supplement they show MUA spike rate in Hz. Given the known variation in firing rates per electrode and the fact that often an electrode can be silent throughout the duration of an experiment, they should show this in a more granular way across the array (not just averaged). They need to be clearer about the spontaneous activity in the organoid before looking at the stimulus-induced activity.

Our response: We thank Reviewer 3 for their encouraging comments and finding our technology of interest. We agree with the reviewer's comment on variation of firing rates. We also sometimes found a channel to be more or less active on different days. We believe this variation could be caused by the cellular reorganization and development of organoids in vivo, differing SNR on different days, or the mice's brain states at the time of the measurements which may have affected neuronal excitability and firing patterns. Altogether, we agree with Reviewer 3 that the averages we presented would be better supported with more granular data. Thus, to address Reviewer 3's first comment, we have added Supplementary Figure 6, showing MUA event rates (Supplementary Figure 6a) for three organoid and three cortex channels along with a raster plot of spontaneous MUA events (Supplementary Figure 6b) for those organoid and cortex channels on recording days 7, 14, 21, 28, 51, and 71.

New Supplementary Figure 6:

Supplementary Figure 6. (a) MUA event rates for three organoid and three cortex channels on recording days 7, 14, 21, 28, 51, and 71 post-implantation. **(b)** Raster plot of spontaneous MUA events in three organoid and three cortex channels.

Comment 2: Fig 3b—this figure is hard to see and appears not to fit the filter definition of crossing their threshold. Many putative MUAs are marked where there appears to be a peak that did not cross the threshold as defined by the red line in the figure. Their pre-defined threshold of -3 to -4 times the SD is somewhat generous—six times is more rigorous and one needs to account for the median noise.

Our response: We thank Reviewer 3 for this comment and opportunity to clarify our figure. In response, we have replaced Figure 3b with data of a finer temporal resolution, so the MUA events are easier to see. The figure now shows 0.25-second-long MUA traces rather than the 10-second-long traces shown before. We agree that the threshold crossings are better seen using this new temporal resolution.

Modified Figure 3b:

(b) Recording of spontaneous MUA from channels O1, O4, C2, and C7 on 21 dpi. Dots indicate MUA events crossing the -3.5 std threshold.

Regarding the threshold, since our recordings were taken from the surface of the brain, the event amplitudes could be smaller compared to that of depth recordings because of the larger distance. Thus our threshold for events is lower compared to penetrating electrode recordings. Additionally, the amplitude of recordings from channels overlaying organoids could be smaller due to the growth of the organoids *in vivo*. As the organoids develop, there may be a smaller density of cells or lower synchrony of cell firing which might lead to the lower MUA amplitudes. Organoids may also not have the recurrent connections of pyramidal cells that contribute to the surface potentials of endogenous mouse cortex. For all these potential reasons, we chose a threshold of 3-4 times the standard deviation, a level which has been used in other high impact studies previously as well [1-3].

The reviewer mentioned accounting for the median noise. Other formative papers also mention the use of median absolute deviation (MAD) and claim that it can be beneficial in the cases where spikes have such large firing rates and amplitudes that they inflate the standard deviation threshold [4-6]. The suggested MAD method sets the spike event threshold to a multiplier times the MAD. We compared this MAD method to our current method of using the *std* and found that the threshold of $6 \cdot \text{MAD}$ yielded similar spike detection results (see below Review Figure 1).

Review Figure 1:

Review Figure 1. Comparison of MUA event detection using thresholds computed using the standard deviation method (red) and median absolute deviation method (blue). X is the data recording. Dots indicate threshold crossings.

Comment 3: *The focus on high gamma LFPs necessarily encounters the difficulty of disentangling high gamma from spikes. The minimal unit on their recordings is 1 sec in Fig 2. Not until we get to figure 3e, do we see the higher time resolution now presented in a different context, i.e. phase locking. We need to see the activity at msec resolution for spiking and we need clarification and more detailed labeling of Fig 2e to see the entire power spectrum and specifically the power bands of interest in relation to the entire spectrum. The Morlet wavelets used by the authors is not a substitute for a Fourier transform or some similar computation.*

Our response: We thank Reviewer 3 for this comment. We agree that high gamma range LFP includes contributions from spiking activity and currently include the below section in our manuscript addressing this concept:

Page 9, Paragraph 1:

“LFP in the high gamma frequency range (>100 Hz) has been suggested to be strongly correlated with local spiking activity (Scheffer-Teixeira et al. 2013; Einevoll et al. 2013; Petterson and Einevoll 2008; Rasch et al. 2008; Ray et al. 2008; Buszaki, Anastassiou, and Koch 2012). In studies with macaque monkeys, high gamma power of LFPs was found to be highly correlated with spiking activity of neurons (Rasch et al. 2008; Berens et al. 2008).”

For that reason, we process LFP and MUA separately and present the data as separate figures (Figures 2 and 3) but do not claim LFP and MUA are completely disentangled. We process low (1-250 Hz) and high (500-3000 Hz) frequency signals separately because LFPs encompass synaptic transmission, action potentials, and intrinsic currents whereas MUA reflects primarily local neuronal firing [7-11] and thus can reveal unique information on the underlying cortex and organoid neural dynamics. For our analysis of phase locking values, where we compare LFP phases with MUA events, we use the theta LFP band to avoid spurious correlations.

In order to address the next part of Reviewer 3’s comment “*we need to see the activity at msec resolution for spiking,*” we performed spike detection by filtering and thresholding, which is the standard way of showing spiking and MUA events in the field. Spectrograms in the high frequency range (>500 Hz) are typically not computed to demonstrate spiking since single-unit spikes and MUA events are too intermittent, leading to no peaks in the power spectrogram. Instead, we plotted detected MUA events with 3-ms time windows in Figure 3c in addition to Figure 3b showing MUA traces for 250 ms duration (see previous comment, figure copied below for reference). For a more detailed view of Figure 3c, we have added Supplementary Figure 7c, showing in greater detail each individual MUA event waveform used for Figure 3c’s averaged waveforms.

Modified Figure 3b:

(b) Recording of spontaneous MUA from channels O1, O4, C2, and C7 on 21 dpi. Dots indicate MUA events crossing the -3.5 sdv threshold.

New Supplementary Figure 7c:

(c) Event-triggered MUA traces shown individually (color traces) and with the average (black traces).

Regarding LFP timescales, in Figure 2, we showed our local field potential (LFP) results and thus used a lower temporal resolution since the signal frequencies of interest (1-250 Hz) and stimulation frequency (2 Hz) were lower. However, we also show a finer temporal resolution in Figure 2d where we illustrate the first LFP response to LED stimuli with a scale bar of 0.2 s. In general, the timescales of our MUA data are shorter than LFP data timescales because MUA events happen in the 0.5-3 kHz range with several-millisecond event durations whereas LFP waveforms last on the order of tens to hundreds of milliseconds.

To address Reviewer 3's comment "*we need clarification and more detailed labeling of Fig 2e to see the entire power spectrum and specifically the power bands of interest in relation to the entire spectrum,*" we have added more numerical labeling to the y axis (Frequency (Hz)) of Figure 2e and modified the subfigure to show the full 1-150 Hz spectrogram for channels 1 and 2. We have also moved the figures showing the 0-10 Hz spectrograms for channels 1 and 2 to Supplementary Figure 4. The modified sentences and figure are shown below. In Figure 2e, we chose to show the power spectrum up to 150 Hz because showing the spectrum above that limit would diminish the bands of interest, namely theta and gamma.

Page 8, Paragraph 2:

"First, we observed increased signal power at the stimulation frequency and its second harmonic, consistent with previous functional activity described in intact cortices of mice (Lopez et al. 2002), cats (Rager and Singer 1998), and humans (Pisarchik, Chholak, and Hramov 2019; Walter, Dovey, and Shipton 1946) (Supplementary Figure 4). Second, visual light stimulation resulted in increases in gamma (30-150 Hz) power, which is consistent with previous work performed in intact visual cortex of mice in vivo (Saleem et al. 2017; Nase et al. 2003)."

Supplementary Figure 4:

Supplementary Figure 4. Spectrograms (0-10 Hz) of the response to 2 Hz, 4 s visual stimulation. Channel 1 is overlaying organoid and Channel 2 is overlaying cortex. The recordings are the same as the ones shown in Figure 2e, acquired in a mouse 69 dpi.

Modified Figure 2e:

(e) Spectrogram (32-150 Hz) of the response (average of 20 trials). Inset: Enlarged spectrogram (1-150 Hz) of the channel overlaying organoid (channel 1) and visual cortex (channel 2). The recording shown in panels b-e was performed 69 dpi.

To address Reviewer 3's comment "*The Morlet wavelets used by the authors is not a substitute for a Fourier transform or some similar computation,*" we use Morlet wavelet analysis because it offers benefits over Fourier-based methods for analyzing non-stationary signals with frequency components that vary in strength over time [12]. A seminal paper on this topic demonstrated that Fourier-, Hilbert-, and Morlet- based methods yield similar results and the parameters chosen are more consequential [13]. To demonstrate the similarity, we have computed the spectrogram of the recording used in Figure 2 using both Fourier- (MATLAB's *pspectrum.m*) and Morlet- based methods. As shown in the figure below (New Supplementary Figure 12), Fourier- and Morlet- based methods yield very similar results.

New Supplementary Figure 12:

Supplementary Figure 12. Spectrograms of the response to light stimuli generated using Morlet- (left) and Fourier- (right) based methods yield similar results. The Short-Time Fourier Transform spectrogram was calculated using MATLAB's *pspectrum.m* function with a 0.2 s time window, 95% overlap, and leak value of 0.85, approximating a Hanning window.

We chose Morlet wavelet analysis for our paper so we could better control the time/frequency tradeoff when analyzing low and high frequency bands. By choosing a lower number of wavelet cycles, we gain better temporal resolution and by choosing a higher number of wavelet cycles, we gain better frequency resolution. Averaging the results of a range of wavelet cycle parameters, we can optimize the time-frequency resolution and gain better resolution throughout the whole spectrogram.

Comment 4: *Given the imprecision of the LFP onset time, computing a 10-20 msec delay in reaching the organoid does not imply coherence of the wave (as they seem to be trying to do). Nor is there a statistic to go along with this very wide range of 10-20 msec to show significance vis à vis the null hypothesis. Thus, this statement is not fully supported: “The consistency in delay times with the literature indicates that the observed responses from the electrode located over the organoid are generated by the activity of human organoid...” {italics are mine} More importantly, concluding that the LFPs came from the organoid is problematic even after they exclude volume conduction. And by the way, there is extensive and extraneous background on volume conduction in the results section. In these small samples, volume conduction is unlikely to have major effects and all of that material fits better as supplemental.*

Our response: We thank Reviewer 3 for this comment and agree that it is difficult to quantify LFP onset time. Instead, in our paper, we quantify the negative peak time and compare it between the electrodes overlaying the visual cortex and organoid. The 10-20 ms delay mentioned in our paper was not our result but the delay between the visual cortex and the RSC region reported in the literature [14]. Our observed delay is 5.7 ms between the furthest organoid and cortical channels. We simply found a propagation pattern of peak times and amplitudes. We apologize for the confusion in our observations versus the literature and we agree with the reviewer that delay analysis alone can suggest but not prove LFP is generated by the organoid. Our paper and revisions in the first rounds clarified that it cannot be due to volume conduction, as requested by Reviewer 1. Extensive and extraneous background on volume conduction was added in response to Reviewer 1’s comments. We revised our sentences on the time delay and added an arrow indicating the 5.7 ms delay in Figure 2d. The revised sentences and figure are as follows:

Page 8, Paragraph 1:

“Peak LFP amplitudes for the channels overlapping with the visual cortex were around 200 μ V and occurred 36 ms after stimulation onset. The channels overlapping with the organoids also detected LFPs with amplitudes of \sim 50 μ V that occurred 41.7 ms after stimulus onset. The 36 ms delay between the stimulation onset and LFP responses measured from the visual cortex channels matches previous observations of evoked response latencies in intact mouse visual cortex (Land et al. 2013; Kuroki et al. 2018; Lopez et al. 2002). The 5.7 ms delay we observed between the furthest organoid and cortical channels ($p < 0.01$, Student’s one-tailed t test, **Supplementary Figure 3**) is on the order of latencies observed between the intact visual cortex and the RSC region in the literature (Funayama et al. 2015). The consistency in our observed delay times from the visual cortex to the organoid with the literature delay times between the visual cortex and RSC could suggest functional connectivity and propagation between the host and transplanted organoid.”

Modified Figure 2d:

(d) Color map indicating peak amplitude (top) and peak delay (bottom) of the response to the first light pulse as shown in panel c. The red dashed lines outline the implantation site, as in panel a. The observed delay between cortex and organoid is 5.7 ms ($p < 0.01$, Student's one-tailed t test).

To address Reviewer 3's comment "Nor is there a statistic to go along with this very wide range of 10-20 msec to show significance vis à vis the null hypothesis," we performed statistical analysis of our observed delay times by shuffling the delay times of each channel, computing the average and standard deviation of delay times, then using Student's one-tailed t test to compute the p-value. The resulting figure showing average delay times is included below (Supplementary Figure 3) for reference and the significance statement has been added to the Figure 2d legend and Methods section:

Modified Figure 2d legend:

"(d) Color map indicating peak amplitude (top) and peak delay (bottom) of the response to the first light pulse as shown in panel c. The red dashed lines outline the implantation site, as in panel a. The observed delay between organoid and cortex is 5.7 ms (arrow, $p < 0.01$, Student's one-tailed t test)."

Methods:

"Statistical significance for delay times was computed by shuffling the 16 channel delay times 1000 times, computing the average and standard deviation of delay times between a channel pairing, then using Student's one-tailed t test to compute the p-value."

New Supplementary Figure 3:

Supplementary Figure 3. Results of Student t test method of shuffling delay data between channels, calculating delays between organoid and cortex channel, and computing the p-value for our observed delay of 5.7 ms ($p < 0.01$).

Comment 5: The basis of the phase locking in the radial histograms does not have statistical support that this reviewer can find and they do not look very convincing. As they noted earlier there are spontaneous LFPs from which the light responsive ones need to be distinguished.

Our response: We have revised the figure 3g-h to clearly show the significance asterisks and made a note in the legend of the statistical test we used. In figure 3g, the asterisk was placed at the preferred angle. Although the radial histograms look spread out, the data during application of stimuli in the channel overlaying visual cortex was computed to be significant using the Rayleigh test for non-uniformity of circular data.

Modified Figure 3g-h:

Modified Figure 3g-h legend:

“(g) Radial histograms of 4.5-Hz LFP signal phases during MUA events in channels 2 and 4. Asterisk indicates $p < 0.05$ using Rayleigh’s test for non-uniformity of circular data and is placed at the preferred angle. (h) Color map of the spatial extent of phase locking; stimulus-induced MUA events of both channels show strongest phase locking to theta oscillations closest to the visual cortex (bottom right). Asterisks indicate $p < 0.05$ between episodes with and without stimulation (only shown on the left plot) using Rayleigh’s test for non-uniformity of circular data.”

Comment 6: *One would like to see controls that change the stimulus duration and intensity and deliver stimulus to ipsilateral eye.*

Our response: We thank Reviewer 3 for this suggestion. We are slightly unclear what the reviewer means by controls as varying the stimulus duration and intensity are expected to yield different responses rather than serve as a control. We used different stimuli parameters other than the 2 Hz shown in this paper to optimize stimuli parameters for our experiments. We show our findings in the below figure. 5 and 10 Hz stimulation yielded clear LFPs in both the organoid and the cortex channels but as the frequency increased above 5 Hz, the train of individual LFP responses to each LED flash of light became less distinct and blended together, potentially either due to signal processing/propagation in the retina, or adaptation in cortex. However, this explanation and analysis was beyond the scope of this paper and will not directly contribute to the main goal and key findings of our paper. Unfortunately, since the experiment concluded more than a year ago, we can no longer run additional experiments to test varying LED intensity or stimulation to the ipsilateral eye but we agree that this is a good suggestion that we will keep in mind for our future experiments.

Review Figure 2:

Review Figure 1. LFP responses to varying stimuli frequencies and durations: 2 Hz for 4 s (top), 5 Hz for 2 s (middle), and 10 Hz for 2 s (bottom). Recordings were taken on 50 dpi.

References

1. Rossant, C., Kadir, S. N., Goodman, D. F., Schulman, J., Hunter, M. L., Saleem, A. B., Grosmark, A., Belluscio, M., Denfield, G. H., Ecker, A. S., Tolias, A. S., Solomon, S., Buzsáki, G., Carandini, M., and K.D. Harris. (2016). 'Spike sorting for large, dense electrode arrays', *Nature neuroscience*, 19(4), 634-641.
2. Bansal, A. K., Truccolo, W., Vargas-Irwin, C. E., & Donoghue, J. P. (2012). 'Decoding 3D reach and grasp from hybrid signals in motor and premotor cortices: spikes, multiunit activity, and local field potentials', *Journal of neurophysiology*, 107(5), 1337-1355.
3. Burns, S. P., Xing, D., & Shapley, R. M. (2010). 'Comparisons of the dynamics of local field potential and multiunit activity signals in macaque visual cortex', *Journal of Neuroscience*, 30(41), 13739-13749.
4. Rey, H. G., Pedreira, C., & Quiroga, R. Q. (2015). 'Past, present and future of spike sorting techniques', *Brain research bulletin*, 119, 106-117.
5. Quiroga, R. Q., Nadasdy, Z., & Ben-Shaul, Y. (2004). Unsupervised spike detection and sorting with wavelets and superparamagnetic clustering. *Neural computation*, 16(8), 1661-1687.
6. Quiroga, R. Q., Reddy, L., Koch, C., & Fried, I. (2007). Decoding visual inputs from multiple neurons in the human temporal lobe. *Journal of neurophysiology*, 98(4), 1997-2007.
7. Supèr, Hans, and Pieter R. Roelfsema. 200'. 'Chronic multiunit recordings in behaving animals: advantages and limitations.' *Progress in Brain Research* (Elsevier).
8. Henrie, J. A., and R. Shapley. 200'. 'LFP power spectra in V1 cortex: the graded effect of stimulus contrast', *J Neurophysiol*, 94: 479-90.
9. Bastos, A. M., and J. M. Schoffelen. 201'. 'A Tutorial Review of Functional Connectivity Analysis Methods and Their Interpretational Pitfalls', *Front Syst Neurosci*, 9: 175.
10. Kajikawa, Y., and C. E. Schroeder. 2011. 'How local is the local field potential?', *Neuron*, 72(5), 847-858.
11. Nason, S. R., A. K. Vaskov, M. S. Willsey, E. J. Welle, H. An, P. P. Vu, A. J. Bullard, C. S. Nu, J. C. Kao, K. V. Shenoy, T. Jang, H. S. Kim, D. Blaauw, P. G. Patil, and C. A. Chestek. 2020. 'A low-power band of neuronal spiking activity dominated by local single units improves the performance of brain-machine interfaces', *Nature biomedical engineering*, 4(10), 973-983.
12. Cohen, M. X. (2019). A better way to define and describe Morlet wavelets for time-frequency analysis. *NeuroImage*, 199, 81-86.
13. Bruns, A. (2004). 'Fourier-, Hilbert- and wavelet-based signal analysis: are they really different approaches?', *Journal of neuroscience methods*, 137(2), 321-332.
14. Funayama, K., G. Minamisawa, N. Matsumoto, H. Ban, A. W. Chan, N. Matsuki, T. H. Murphy, and Y. Ikegaya. 2015. 'Neocortical Rebound Depolarization Enhances Visual Perception', *PLoS Biol*, 13: e1002231.

REVIEWER COMMENTS

Reviewer #3 (Remarks to the Author):

They did an excellent job of responding and I believe the paper has satisfied all of my criticisms.

REVIEWERS' COMMENTS

Reviewer #3 (Remarks to the Author):

“They did an excellent job of responding and I believe the paper has satisfied all of my criticisms.”

Our reply: *We are glad we satisfied all reviewer comments and thank the reviewers for their input and the opportunity to strengthen our overall manuscript.*